



# A circulation-based performance atlas of the CMIP5 and 6 models for regional climate studies in the northern hemisphere

Swen Brands[1,2]

[1]MeteoGalicia, Consellería de Medio Ambiente, Territorio y Vivienda - Xunta de Galicia, Santiago de Compostela, Spain
[2]Tragsatec, Santiago de Compostela, Spain

**Correspondence:** Swen Brands (swen.brands@gmail.com)

**Abstract.** Global Climate Models are a keystone of modern climate research. In many applications relevant for decision making, and particularly when deriving future projections with the delta-change method, they are assumed to be perfect. However, these models have not been originally developed to reproduce the regional-scale climate, which is where information is needed in practice. To overcome this dilemma, two general efforts have been made since their introduction in the late

1960ies. First, the models themselves have been steadily improved in terms of physical and chemical processes, parametrization schemes, resolution and complexity, giving rise to the term "Earth System Model". Second, the global models' output has been refined at the regional scale using Limited Area Models or statistical methods in what is known as dynamical or statistical *downscaling*. Both approaches, however, are in principle unable to correct errors resulting from a wrong representation of the large-scale circulation in the global model. Also, dynamical downscaling has a high computational demand and thus cannot

be applied to all available global models in practice. On this background, there is an ongoing debate in the downscaling community on whether to thrive away from the "model democracy" paradigm towards a careful selection strategy based on the global models' capacity to reproduce key aspects of the observed climate. The present study attempts to be useful for such a selection by providing a performance assessment of the historical global model experiments from CMIP5 and 6 based on recurring regional atmospheric circulation patterns (Lamb, 1972). The latest model generation is found to perform better on

average, which can be partly explained by a moderately strong statistical relationship between performance and *horizontal* resolution in the atmosphere. A few models rank favourably over almost the *entire* northern hemisphere extratropics, but the better models tend to be less complex than others. Model selection should therefore not solely rely on model performance but also on model complexity and a discussion is needed on how to combine these two criteria. Internal model variability only has a small influence on the model ranks. Reanalysis uncertainty is an issue in Greenland and the surrounding seas, the southwestern

United States and the Gobi desert, but is otherwise negligible.

## 1   Introduction

*General Circulation Models* (GCMs) are numerical models capable to simulate the temporal evolution of the global atmosphere or ocean. This is done by integrating the equations describing the conservation laws of physics along time as a function of varying forcing agents, starting with some initial conditions (AMS, 2020). If run in standalone mode, an Atmospheric General





Circulation Model (AGCM) is coupled with an indispensable land-surface model (LSM) only, whilst the remaining components of the extended climate system (also called "realms" in the nomenclature of the Earth System Grid Federation), including ocean, sea-ice and vegetation dynamics (depending on the model also atmospheric chemistry, aerosols, ocean biogeochemistry and even ice-sheet dynamics) are read-in from static datasets instead of being simulated online (Gates, 1992; Eyring et al., 2016; Waliser et al., 2020). In these "atmosphere-only" experiments, the number of coupled realms is kept at a minimum in

order to either isolate the sole atmospheric response to temporal variations in the aforementioned other components (Schubert et al., 2016; Brands, 2017; Deser et al., 2017) or to put all available computational resources into the proper simulation of the atmosphere, e.g. by augmenting the spatial and temporal resolution (Haarsma et al., 2016). This kind of experiment is traditionally hosted by the Atmospheric Model Intercomparison Project (AMIP) (Gates, 1992).

     In a *Global Climate Model*, interactions and feedbacks between the aforementioned realms are explicitly taken into ac-

count by coupling the AGCM and LSM with other component models. In the "ocean-atmosphere" configuration (AOGCM, for Atmosphere-Ocean General Circulation Model), the AGCM plus LSM are coupled with an ocean general circulation model (OGCM) and a sea-ice model. Further model components representing the effects of vegetation, atmospheric chemistry, aerosols, ocean biogeochemistry and even ice-sheet dynamics are then optionally included with the final aim to reach a representation of the climate system as comprehensive as possible with the current level of knowledge and available computational

resources. In this context, a model capable to resolve both the terrestrial and oceanic processes affecting the global carbon cycle is commonly referred to as "Earth System Model" (ESM) (Collins et al., 2011). Hence, while an AOGCM is already complex, an ESM is even more so and thus more prone to error sources and model uncertainties, making it difficult to directly compare the simulated climate with the observed one (Watanabe et al., 2011; Yukimoto et al., 2011).

     Since these *coupled* model experiments are the best known approximation to the real climate system, they constitute the start-

ing point of most climate change impact-, attribution- and mitigation studies. For use in impact studies, the coarse-resolution GCM output is usually downscaled with statistical or numerical models (Maraun et al., 2010; Jacob et al., 2014; Gutiérrez et al., 2013; San-Martín et al., 2016) or a combination thereof (Turco et al., 2011), in order to provide information on the regional to local scale where it can then be used for decision making.

     Now while downscaling methods are able to imprint the effects of the local climate factors on the coarse resolution GCM,

they do not correct errors inherited from a wrong representation of the large-scale atmospheric circulation (Prein et al., 2019). Hence, the only physically consistent way to circumvent this "circulation error" is choosing a GCM (or group of GCMs) capable to realistically simulate the climatological statistics of the regional-scale circulation. This is why careful GCM selection for long has been the subject of any careful downscaling approach applied in a climate change context (Hulme et al., 1993; Mearns et al., 2003; Brands et al., 2013). However, due to the availability of many GCMs from many different groups, this

idea has been partly replaced by the "model democracy" paradigm discussed e.g. in Knutti et al. (2017), where as many GCMs as possible are applied irrespective of their performance in present-day conditions (Jacob et al., 2014). In the recent past, the importance of careful model selection has been re-emphasized in the context of bias correction, which can be considered a special case of statistical downscaling (Maraun et al., 2017). It should be also remembered that GCMs by definition were not developed to realistically represent regional-scale climate features (Grotch and MacCracken, 1991; Palmer and Stevens,





2019) and that they have been pressed into this role during the last 3 decades due to the ever increasing demand for climate information on this scale. Hence, finding a GCM capable to reproduce the regional atmospheric circulation in a systematic way, i.e. in many regions of the world, would be anything but expected.

In the present study, a total of 116 historical runs from 46 distinct GCMs (or GCM versions) of the fifth and sixth phase of the Coupled Model Intercomparison Project (CMIP5 and 6) are evaluated in terms of their capability to represent the present-day climatology of the regional atmospheric circulation as represented by the frequency of the 27 circulation types proposed by Lamb (1972). Based on the proposal in Jones et al. (2013) that this scheme can in principle be applied within a latitudinal band from 30°N to 70°N, it is here used with a sliding coordinate system (Otero et al., 2017) running along the grid-boxes of a 2.5° latitude-longitude grid covering the entire Northern Hemisphere mid-latitudes.

In Section 2 and 3, the applied data, methods and software are described. In Section 4, the results of an *overall* model
performance analysis including all 27 circulation types are presented. First, the three aforementioned regions are identified where reanalysis uncertainty might compromise the results of any GCM performance assessment based on a single reanalysis. Then, an atlas of *overall* model performance is provided for each participating model (Sections 4.1 to 4.7). The present article file focusses on the evaluation w.r.t. ERA-Interim, complemented by pointing out deviations from the evaluation w.r.t. JRA-55 in the 3 relevant regions. The *full* atlas of the evaluation against JRA-55 is provided in the supplementary material to this
study (see "figs-refjra55" folder therein). In Section 4.8, the atlas is summarized and the role of *internal* model variability assessed with 70 additional historical runs from a subgroup of 12 models. Finally, the results of a *specific* model performance evaluation for each circulation type are provided in Section 5, followed by a discussion of the main results and some concluding remarks in Section 6. For the sake of simplicity, the model performance atlas is grouped by the geographical location of the coupled models' *coordinating* institutions, having in mind that most model developments are actually international or even
transcontinental collaborating efforts.

## 2   Applied Data and Usage

The study resides on *6-hourly instantaneous* sea-level pressure (SLP) model data retrieved from the Earth System Grid Federation (ESGF) data portals (e.g. https://esgf-data.dkrz.de/projects/esgf-dkrz/) whose Digital Object Identifiers (DOIs) can be obtained following the references in Table 1. These model runs are evaluated against reanalysis data from ECMWF
ERA-Interim (Dee et al., 2011) (https://apps.ecmwf.int/datasets/data/interim-full-daily/levtype=sfc/) and Japan Meteorological Agency (JMA) JRA-55 (Kobayashi et al., 2015) (https://rda.ucar.edu/datasets/ds628.0/, DOI:10.5065/D6HH6H41). In a first step, and in order to compare as many distinct models as possible, a single historical run was downloaded for each model for which the aforementioned data were available for the 1979-2005 period. If several historical integrations for given model version were available, then the first member was chosen. In Section 4.8, it will be shown that the selection of alternative
members from a given ensemble does *not* lead to substantial changes in the results. Out of the 25 models used in CMIP6, 20 were run with the "f1", four with the "f2" and one with the "f3" forcing datasets (Eyring et al., 2016) (see Table 1). Not only version *pairs* from CMIP5 to CMIP6 are considered, but also model versions either not having a predecessor in CMIP5





or a successor in CMIP6. In the most favourable case, two versions of a given model are available for both CMIP5 and 6: A higher-resolution configuration considering fewer realms (the AOGCM configuration), complemented by a lower-resolution

setup including more realms, ideally reaching the status of an ESM.

An overview of the 46 applied model versions is provide in Table 1. The table provides information about the component AGCMs and OGCMs , their horizontal and vertical resolution, details about the considered runs and the reference publications. The models' degree of complexity, here defined by the number of considered realms, is also indicated. "AOGCM" refers to the basic coupling configuration covering atmosphere, land-surface, ocean and sea-ice dynamics only. As a trade-off between

a fully comprehensive ESM, as described e.g. in Yukimoto et al. (2011) for the case of MRI-ESM, and the considerations of other model developers, a model is here declared an ESM if both the terrestrial *and* oceanic biogeochemical processes relevant for the carbon cycle are calculated online instead of being read-in from external files, which is in line with the proposal made in Collins et al. (2011). Assignment to either of the two groups was made on the basis of the reference articles and the metadata provided the model output files. The "source" attribute present in these files contains the specifications of the individual model

components. This attribute was extracted for each model and permanently stored at https://doi.org/10.5281/zenodo.4452081. Since many models are ahead of the basic AOGCM configuration but yet cannot be considered ESMs because terrestrial and/or ocean biogeochemistry processes are missing, these additional components are added to the base category "AOGCM" (plus either "tbgc" or "obgc", respectively). Likewise, if a given model is more complex than the basic ESM configuration because atmospheric (photo)chemistry, aerosols and/or ice sheet dynamics are also taken into account, this is also indicated (ESM

plus "chem", "aero" and/or "icesheet", respectively). Note that a model's complexity is not solely defined by the number of considered realms but also the by the variety of coupled processes. For these coupling details, the interested reader is referred to the reference articles listed in Table 1.

For 12 selected models (ACESS-ESM1, HadGEM2-ES, EC-Earth3, IPSl-CM5A-LR, IPSL-CM6A-LR, MIROC-ES2L, MPI-ESM1-2-LR, MPI-ESM1-2-HR, MRI-ESM2, NorESM2-LM, NorESM2-MM, NESM3), a total of 70 additional histori-

cal integrations (between 1 and 17 additional runs per model) were retrieved from the respective ensembles in order to assess the effects of internal model variability. By definition of the experimental protocol followed in CMIP, ensemble spread relies on initialization from distinct starting dates of the corresponding pre-industrial control runs (or similar, shorter runs as e.g. indicated in Roberts et al. (2019)), i.e. on "initial conditions uncertainty" (Stainforth et al., 2007).

## 3 Methods

### 3.1 Lamb Weather Types

The classification scheme used here is based on H.H. Lamb's practical experience when grouping daily instantaneous SLP maps for the British Isles and interpreting their relationships with the regional weather (Lamb, 1972). His subjective classification scheme contained 27 classes and was brought to an automated and objective approach by Jenkinson and Collison (1977) in what is known as the "Lamb Circulation Types" or "Lamb Weather Types" (LWTs) approach (Jones et al., 1993, 2013).





The spatial extension of the 16-point coordinate system defining this classification is 30 longitudes $\times$ 20 latitudes with longitudinal and latitudinal increments of 10° and 5°, respectively (see Figure 1 for an example over the Iberian Peninsula). The following numbers are place-holders of instantaneous SLP values (in hPa) at the corresponding location $p$ (from West to East and North to South):


$$p01 \quad p02$$

$$p03 \quad p04 \quad p05 \quad p06$$

$$p07 \quad p08 \quad p09 \quad p10$$

$$p11 \quad p12 \quad p13 \quad p14$$

$$p15 \quad p16$$

, and the variables needed for classification are defined as follows:

$$Westerly\ flow\ (W) \ = \ \frac{1}{2}(p12+p13) - \frac{1}{2}(p04+p05) \tag{1}$$

$$Southerly\ flow\ (S) \ = \ a\,[\frac{1}{4}(p05+2\times p09+p13) - \frac{1}{4}(p04+2\times p08+p12)] \tag{2}$$

$$Resulting\ flow\ (F) \ = (S^2+W^2)^{1/2} \tag{3}$$

$$Westerly\ shear\ voriticity\ (ZW) \ = \ b\,[\frac{1}{2}(p15+p16) \ - \ \frac{1}{2}(p08+p09)]$$
$$- \ c\,[\frac{1}{2}(p08+p09) \ - \ \frac{1}{2}(p01+p02)]$$

$$\tag{4}$$

$$Southerly\ shear\ voriticity\ (ZS) \ = \ d\,[\frac{1}{4}(p06+2\times p10+p14)$$
$$-\frac{1}{4}(p05+2\times p09+p13)$$
$$-\frac{1}{4}(p04+2\times p08+p12)$$
$$+\frac{1}{2}(p03+2\times p07+p11)]$$

$$\tag{5}$$


where $a = 1/cos(\phi)$, $b = sin(\phi)/sin(\phi - \delta\phi)$, $c = sin(\phi)/sin(\phi + \delta\phi)$ and $d = 0.5(cos(\phi)^2)$; $\phi$ is the central latitude and $\delta\phi$ is the latitudinal distance.

The 27 classes are then defined following Jones et al. (1993) and Jones et al. (2013):

1. The direction of flow is $\tan^{-1}(W/S)$. Add $180°$ if $W$ is positive. The appropriate direction is calculated on an eight-point compass allowing $45°$ per sector. Thus, as an example, a westerly flow would occur between $247.5°$ and $292.5°$.

2. If $|Z|$ is less than $F$, then the flow is essentially straight and corresponds to one of the 8 purely directional types defined by Lamb: Northeast (NE), East (E), SE, S, SW, W, NW, N.

3. If $|Z|$ is greater than $2F$, then the pattern is either strongly cyclonic (for $Z > 0$) or anticyclonic (for $Z < 0$), which corresponds to Lamb's pure cyclonic (PC) or anticyclonic type (PA), respectively.

4. If $|Z|$ lies between F and 2F, then the flow is partly directional and either cyclonic or anticyclonic, corresponding to Lamb's *hybrid* types. There are 8 directional-*anticyclonic* types (Anticyclonic Northeast (ANE), Anticyclonic East (AE), ASE, AS, ASW, AW, ANW, AN and another 8 directional-*cyclonic* types (Cyclonic Northeast (CNE), Cyclonic East (CE), CSE, CS, CSW, CW, CWN, CN.

5. If $F$ is less than 6 and $|Z|$ is less than 6, there is light indeterminate flow corresponding to Lamb's unclassified type $U$. The choice of 6 is dependent on the grid spacing and would need tuning if used with a finer grid resolution.

An illustrative example for the results obtained from this scheme is provided in Figure 1 for the case of the central Iberian Peninsula. Shown is the coordinate system and the composite SLP maps for a subset of 14 LWTs, as well as the respective relative occurrence frequencies, taken from Brands et al. (2014) (courtesy to John Wiley and Sons, Inc.).

Particularly since the 1990s, this classification scheme has been used in many other regions of the NH mid-latitudes (Trigo and DaCamara, 2000; Spellman, 2016; Wang et al., 2017; Soares et al., 2019). Since the LWTs are closely related to the local-scale variability of virtually all meteorological- and many other environmental variables (Lorenzo et al., 2008; Wilby and Quinn, 2013), they constitute an *overarching* concept to verify GCM performance in present climate conditions and have been used so in a number of studies (Hulme et al., 1993; Osborn et al., 1999; Otero et al., 2017).

Here, for each model run and the ERA-Interim or JRA-55 reanalysis, the 6-hourly instantaneous SLP data from 01/01/1979 to 31/12/2005 are bi-linearly interpolated to a regular latitude-longitude grid with a resolution of $2.5°$. Then, the Lamb classification scheme is applied for each time instance and grid-box, using a sliding coordinate system whose centre is displaced from one grid-box to another in a loop recurring all latitudes and longitudes of the aforementioned grid within a band from 35 to 70°N. Note that the geographical domain is cut at 35°N (and not at 30°) because the various available reanalyses are known to produce comparatively large differences in their estimates for the "true" atmosphere when approaching the tropics (Brands et al., 2012, 2013). Also, since some models do not apply the Gregorian calendar but work with 365 or even 360 days per year, *relative* instead of absolute LWT frequencies are considered. Further, since HadGEM2-CC and HadGEM2-ES lack SLP data for December 2005, this month is equally dropped from ERA-Interim or JRA-55 when compared with these models.





As mentioned above, the LWT approach has been successfully applied for many climatic regimes of the NH, including the extremely continental climate of central Asia (Wang et al., 2017), which confirms the proposal made in Jones et al. (2013) that the method in principle can be applied in a latitudinal band from 30 to 70°N. Here a criterion is introduced to explicitly test this assumption. Namely, it is established that LWTs cannot not be used at a given grid-box if the relative frequency for any of the 27 types is lower than $0.1\%$ percent (i.e. 15 annual occurrences on average). Note that, already in its original formulation

for the British Isles, some LWTs were found to occur with relative frequencies as small as $0.47\%$ (Perry and Mayes, 1998). This is why the $0.1\%$ threshold seems reasonable in the present study. If at a given grid-box this criterion is not met in the LWT catalogue derived from ERA-Interim or alternatively JRA-55, then this grid-box does not participate in the evaluation.

### 3.2    Applied GCM performance measures

To measure GCM performance, the Mean Absolute Error (MAE) of the $n = 27$ relative LWT frequencies obtained from a

given model (m) w.r.t. to those obtained from the reanalysis (o) are calculated at a given grid-box:

$$MAE = \frac{1}{n}\Sigma_{i=1}^{n}|m_i - o_i| \tag{6}$$

The MAE is then used to rank the 46 distinct models at this grid-box. The lower the MAE, the lower the rank and the better the model. After repeating this method for each grid-box of the NH, both the MAE values and ranks are plotted for each individual model on a polar stereographic projection.

In addition to the MAE measuring *overall* performance, the *specific* model performance for each LWT is also assessed. This is done because, by definition of the MAE, errors occurring in the more frequent LWTs are penalized more than those occurring in the rare LWTs. Hence, a low MAE might mask errors in the least frequent LWTs. For a LWT-specific evaluation, the simulated frequency map for a given LWT and model are compared with the corresponding map from the reanalysis by means of the Taylor Diagram (Taylor, 2001). This diagram compares the spatial correspondence of the simulated and observed

(or "quasi-observed" since *reanalysis* data are used) frequency patterns by means of 3 complementary statistics. These are the Pearson correlation coefficient ($r$), the standard deviation ratio ($ratio = \sigma_m/\sigma_o$), with $\sigma_m$ and $\sigma_o$ being the the standard deviation of modelled and observed frequency patterns, and the normalized centred root mean-square error (CRMSE):

$$CRMSE = \frac{\sqrt{\frac{1}{n}\sum_{i=1}^{n}(cm_i - co_i)^2}}{\sigma_o} \tag{7}$$

, with $n = 2016$ grid-boxes covering the NH mid-latitudes and $cm$ and $co$ the modelled and observed frequency patterns after

subtracting their own mean value (i.e. both the minuend and subtrahend are anomaly fields, "c" refers to centred). Normalization enables for comparison with other studies using the same method.





## 3.3 Applied *Python* packages

The coding to the present study relies on the *Python v2.7.13* packages *xarray v0.9.1* written by Hoyer and Hamman (2017) (https://doi.org/10.5281/zenodo.264282), *NumPy v1.11.3* written by Harris et al. (2020) (https://github.com/numpy/numpy),
*Pandas v0.19.2* written by McKinney (2010) (https://doi.org/10.5281/zenodo.3509134) and *SciPy v0.18.11* written by Virtanen et al. (2020) (https://doi.org/10.5281/zenodo.154391); here used for i/o tasks and statistical analyses. The *Matplotlib v2.0.0* package written by Hunter (2007) (https://doi.org/10.5281/zenodo.248351), as well as the Basemap v1.0.7 toolkit (https://github.com/matplotlib/basemap) are applied for plotting and the functions written by Gourgue (2020) (https://doi.org/10.5281/zenodo.3715535) for generating Taylor diagrams.

## 210 4 Overall model performance results

In Figure 2, the MAE of JRA-55 w.r.t. ERA-Interim is mapped (panel a), complemented by the corresponding rank within the multi-model ensemble plus JRA-55 (panel b). In the ideal case, the MAE for JRA-55 is lower than for any of the 46 CMIP models, which means that the alternative reanalysis ranks first and that a change in the reference reanalysis does not influence the model ranking. This result is indeed obtained for a large fraction of the NH. However, in the Gobi desert, in Greenland and
the surrounding seas, and particularly in the southwestern United States of America, substantial differences are found between the two reanalyses. Since different reanalyses from roughly the same generation are in principle equally representative of the "truth" (Sterl, 2004), the models are here evaluated twice in order to obtain a robust picture of their performance. In the present article file, the evaluation results w.r.t. to ERA-Interim are mapped and deviations from the evaluation against JRA-55 in the 3 relevant regions are pointed out in the text. In the remaining regions, reanalysis uncertainty plays a minor role.
Nevertheless, for the sake of completeness, the interested reader can see in the full atlas of the JRA-55-based evaluation in the supplementary material to this study. For a quick overview of the results, Table 1 indicates whether a given model closer agrees with ERA-Interim or JRA-55 in the 3 sensitive regions. In the following, this is referred to as "reanalysis affinity".

Figure 2 also shows that the LWT usage criterion defined in Section 3.1 is met almost everywhere in the domain, except in the high-mountain areas of central Asia (grey areas within the performance maps indicate that the criterion is not met). This
region is governed by the monsoon rather than the turnover of dynamic low- and high pressure systems the LWT approach was developed for. It is thus justified to use the approach over such a large domain.

Grouped by their geographical origin, Sections 4.1 to 4.7 describe how the 46 participating coupled models are composed in terms of their atmosphere, land-surface and ocean models (and others) in order to make clear whether there are shared components between nominally different models that might explain common error structures. Then, the regional error and
ranking details are provided. In Section 4.8, these results are summarized in a single boxplot and put into relation with the resolution setup of the atmosphere and ocean component models. The role of internal model variability is also assessed there.

The first result common to all models is the spatial structure of the absolute error expressed by the MAE. Namely, the models tend to perform better over ocean areas than over land and perform poorest over high-mountain areas, particularly in central Asia. Further regional details are documented in the following sections.





### 4.1 Model contributions from the United Kingdom and Australia

All components of the Hadley Centre Global Environment Model version 2 (HadGEM2) have been developed independently by the *Met Office Hadley Centre* during the last decades. Atmospheric, land-surface and ocean dynamics are represented by the HadGAM2, MOSES2 and HadGOM2 models, respectively. Both the terrestrial and ocean carbon cycles are taken into account by the two HadGEM2 versions considered here (CC and ES), which only differ by the inclusion of gas-phase chemistry in the therefore slightly more complete ES version (Collins et al., 2011; Martin et al., 2011). This centre's model contributions to CMIP6 are following the concept of seamless prediction (Palmer et al., 2008), in which lessons learned from short-term numerical weather forecasting are exploited for the improvement of longer-term predictions/projections up to climatic time-scales, using a "unified" or "joint" model for all purposes. For atmosphere and land-surface processes, these are the Unified Model Global Atmosphere 7 (UM-GA7) AGCM and the Joint UK Land Environment Simulator (JULES) (Walters et al., 2019). However, the specific CMIP6 model version considered here (HadGEM3-GC31-MM) is a very high-resolution AOGCM configuration with the ocean biogeochemistry module turned off (Roberts et al., 2019). Unlike HadGEM2-ES and CC, HadGEM3-GC31-MM is therefore not considered an ESM.

With nearly identical error and ranking patterns associated with the aforementioned almost identical configuration, already the two model versions used in CMIP5 (HadGEM2-CC and ES) yield a good to very good performance which, for the European sector, is in line with Perez et al. (2014) and Stryhal and Huth (2018). Only a close look reveals slightly lower errors for the ES version, particularly in a region extending from western France to the Ural mountains (see Figure 3). Both CMIP5 versions are outperformed by HadGEM3-GC31-MM. While HadGEM2-CC and ES rank very well in Europe and the central North Pacific only, HadGEM3-GC31-MM does so in virtually all regions of the NH mid-latitudes except in central Asia. It is undoubtedly one of the best models considered here. However, unlike its CMIP5 predecessors, it is not an ESM.

While CSIRO-MK (not assessed here) was an independently developed model of the Australian research community (Collier et al., 2011), the *Community Climate and Earth System Simulator* (ACCESS) depends to a large degree on the aforementioned models from the Met Office Hadley Centre. ACCESS1.0, the starting point for the new Australian ESM, makes use of the same atmosphere and land-surface components as HadGEM2 (see above), but is run in AOGCM mode only. As such, it is considered the "control" configuration of all further developments towards the ESM configuration made by the Australian research community (Bi et al., 2013). ACCESS1.3 is the first step into this direction. Instead of HadGAM2, it uses a slightly modified version of the Met Office Global Atmosphere 1.0 (GA1) AGCM coupled with the CABLE1.8 land surface model developed by CSIRO. ACCESS-CM2 is the AOGCM version used in CMIP6, relying on the UM10.6-GA7.1 AGCM (also used in HadGEM3-GC31-MM) coupled with CABLE2.5 (Bi et al., 2020). ACCESS-CM2, however, was run with a lower horizontal resolution in the atmosphere than HadGEM3-GC31-MM. ESM status is finally attained by ACCESS-ESM1.5; at the expense of using somewhat older AGCM and LSM versions (UM7.3-GA1 and CABLE2.4) than in ACCESS-CM2, probably in order to free computational resources for the ocean biogeochemistry model WOMBAT (Ziehn et al., 2020). With GFDL-MOM and CICE, all ACCESS models use the same ocean and sea-ice models, which differ from those used in the HadGEM model family. The OASIS coupler (Valcke, 2006) is again applied by both model families.





Within the ACCESS model family, version 1.0 performs best (see Figure 3). The corresponding error and ranking patterns
are virtually identical to HadGEM2-ES and HadGEM2-CC, which is due to the same AGCM used in these three models
(HadGAM2). The 3 more independent versions of ACCESS (1.3, CM2 and ESM1.5) roughly share the same error pattern,
which differs from ACCESS1.0 in some regions. While the 3 independent developments perform worse in the North Atlantic
and western North Pacific, they do better in the eastern North Pacific off the coast of Japan and, in case of ACCESS-CM2, also
in the high mountain areas of central Asia and the Mediterranean. In the latter two regions, the performance of ACCESS-CM2
is comparable to HadGEM3-GC31-MM. Overall, version 1.0 performs best within the ACCESS model family.

The two HadGEM2 versions and also ACCESS1.3 compare better with JRA-55 in the southwestern U.S. but thrive towards
ERA-Interim in the seas around Greenland and in the Gobi desert. HadGEM3-GC31-MM, ACCESS1.0, ACCESS-CM2 and
ACCESS-ESM1.5 have similar reanalysis affinities, except for thriving towards JRA-55 in the seas around Greenland and
for showing virtually no sensitivity in the Gobi desert in case of the ACESS-ESM1.5 (compare Figure 3 with the "figs-
refjra55/maps/rank" folder in the supplementary material).

### 4.2 Model contributions from North America

Figure 4 shows the respective results for the models developed in North America. Each of the four model families are built
upon independent and long-standing research lines.

From model version CM3 to CM4, the *Geophysical Fluid Dynamics Laboratory* (GFDL) has updated and considerably
increased the resolution of its in-house AGCM and OGCM (Griffies et al., 2011; Held et al., 2019) —except for the number of
vertical levels in the AGCM, which was reduced from CM3 to CM4—, and this actually pays off in terms of model performance
(see Figure 4). While GFDL-CM3 only ranks well in an area ranging from the Great Plains to the central North Pacific, GFDL-
CM4 yields balanced results over the entire NH and is one of the best models considered here. Notably, GFDL-CM4 also
performs well over central Asia and in an area ranging from the Black Sea to the Middle East, which is where most of the other
models perform less favourable. Note also that GFDL's Modular Ocean Model (MOM) is the standard OGCM in all ACCESS
models and is also being used in BCC-CSM2-MR (see Table 1 for details).

The *Goddard Institute of Space Studies* model versions used in CMIP5 are AOGCMs including the effects of atmospheric
chemistry and aerosols. The two versions are identical except for the ocean component: HYCOM was used in GISS-E2-H and
Russel Ocean in GISS-E2-R (Schmidt et al., 2014). Russel Ocean was then developed to GISS Ocean v1 for use in GISS-E2.1-
G (Kelley et al., 2020), the CMIP6 model version assessed here, which however was run without the aforementioned chemistry
and aerosol modules (note that the 6-hourly SLP data for the more complex model versions contributing to CMIP6 were not
available from the ESGF data portals). All these versions comprise a relatively modest resolution for the atmosphere and ocean
and no refinement was undertaken from CMIP5 to 6. However, many parametrization schemes were improved. GISS-E2.1-G
generally ranks better than its predecessors, except in eastern Siberia and China, where very good ranks are obtained by the
two CMIP5 versions (see Figure 4). The small differences between the results for GISS-E2-H and R might stem from internal
model variability (see also Section 4.8) or indeed from the use of two distinct OGCMs. Unfortunately, all versions of the GISS





model are plagued by pronounced performance differences from one region to another, meaning that they are less balanced than e.g. GFDL-CM4.

The *National Center for Atmospheric Research* (NCAR) Community Climate System Model 4 (CCSM4) is composed of
the Community Atmosphere and Land Models (CAM and CLM), the Parallel Ocean Program (POP) and the los Alamos Sea Ice Model (CICE), combined with the CPL7 coupler (Gent et al., 2011; Craig et al., 2012). The CMIP5 experiment considered here was run with a classical AOGCM configuration. During the course of the last decade, CCSM4 has been further developed to CESM1 and 2 (Hurrell et al., 2013; Danabasoglu et al., 2020) which, due to data availability issues, can unfortunately not be assessed here (the respective data for CESM2 are available, but only for 15 out of the 27 years considered here). However,
CMCC-CM2 and NorESM2 are almost entirely made up by components from CESM1 and 2, respectively, and should thus be also indicative for the performance of the latter (see Sections 4.5 and 4.6). Similar data availability problems apply to CanESM5 (Swart et al., 2019), the latest ESM generation contributed by the *Canadian Centre for Climate Modeling and Analysis* (CCCma). Hence, only CanESM2 (Chylek et al., 2011) —the CMIP5 antecessor— can be assessed here.

Results indicate a comparatively poor performance for both CCSM4 and CanESM2. Exceptions are found along the North
American west coast and the Labrador Sea, where both models perform well; in the central to eastern subtropical Pacific and in northwestern Russia plus Finland, where CCSM4 performs well; and in Quebec, Scandinavia and eastern Siberian, where CanESM2 ranks well (see Figure 4). As for the GISS models, both CCSM4 and CanESM2 are also plagued by large regional performance differences.

Regarding the models' reanalysis affinity, GFDL-CM3 thrives towards ERA-Interim in the seas around Greenland and
towards JRA-55 in the Gobi desert, while being almost insensitive to reanalysis choice in the southwestern United States (compare Figure 4 with the "figs-refjra55/maps/rank" folder in the supplementary material to this article). GFDL-CM4 has similar reanalysis affinities, but largely improves (by up to 20 ranks) in the southwestern United States when evaluated against JRA-55. Results for GISS-E2-H and GISS-E2-R are slightly closer to ERA-Interim in the southwestern U.S. and otherwise virtually insensitive to reanalysis choice. GISS-E2-1-G is virtually insensitive in all 3 regions. CanESM2 ranks consistently
better if compared with JRA-55, with a stunning improvement of up to 30 ranks in the southwestern United States, and CCSM4 slightly thrives towards ERA-Interim in all 3 regions.

### 4.3 Model contributions from France

The CMIP5 contributions from the *Centre National de Recherches Météorologique (CNRM)* and *Institut Pierre-Simon Laplace (IPSL)* use the same OGCM and coupler, i.e. the Nucleus for European Modelling of the Ocean model (NEMO) (Madec et al.,
1998; Madec, 2008) and OASIS, but differ in their remaining components. CNRM-CM5 comprises the ARPEGE AGCM, ISBA land-surface model and GELATO sea-ice model (Voldoire et al., 2013) whereas IPSL makes use of LMDZ, ORCHIDEE and LIM, respectively (Dufresne et al., 2013). For CNRM-CM6-1, these components were updated and an atmospheric chemistry model was implemented in addition (Voldoire et al., 2019). Note that all CNRM model versions except CNRM-ESM2-1 (Séférian et al., 2019) are AOGCMs (plus interactive atmospheric chemistry in CNRM-CM6-1 and CNRM-CM6-1-HR)





whereas all model versions from IPSL are considered ESMs (see Table 1). Consequently, the CNRM models could be gener-
ally run with a much finer resolution in the atmosphere and ocean than the more complex IPSL models.

Within the CNRM model family, CNRM-CM5 is found to perform very well except in the central North Pacific, the southern
USA and in a subpolar belt extending from Baffinland in the West to western Russia in the East (see Figure 5). This includes a
good performance over the Rocky Mountains and central Asia. From CNRM-CM5 to CNRM-CM6-1, performance gains are
obtained in the central North Pacific, the southern USA, Scandinavia and western Russia which, however, are compensated by
performance losses in the entire eastern North Atlantic and in an area covering Manchuria, Korea and Japan. A similar picture
is obtained for CNRM-ESM2-1 whereas a performance *loss* is observed for for CNRM-CM6-1-HR. This is surprising since, in
addition to improved parametrization schemes, the model resolution in the atmosphere and ocean was particularly increased in
the latter model version. Under these circumstances, CNRM-CM6-1-HR is actually the only model suffering clear performance
*losses* from CMIP5 to 6. The reasons for this are unknown and should be assessed in future studies.

While missing in all CNRM model versions except CNRM-ESM2-1, the ocean carbon-cycle was an integral part of the IPSL
model already in CMIP5 (Dufresne et al., 2013) and the associated computational costs likely might have forced this group to
run their model versions with a modest resolution in the atmosphere (LMDZ) and ocean (NEMO). This changed for the better
with IPSL-CM6A-LR, where a more competitive resolution was applied and all component models were improved (Boucher
et al., 2020; Hourdin et al., 2020). The result is a considerable performance increase from CMIP5 to CMIP6. Whereas both
IPSL-CM5A-LR and IPSL-CM5A-MR perform poorly, IPSL-CM6A-LR does much better virtually *anywhere* in the NH, a
results that is virtually insensitive to the effects of internal model variability arising from initial conditions uncertainty (see
Section 4.8).

The quite different results between the CNRM and IPSL models indicate that the common ocean component (NEMO) only
marginally affects the simulated atmospheric circulation as defined here. All CNRM models, and also IPSL-CM6A-LR, thrive
towards Interim in the southwestern U.S. and towards JRA-55 in the seas around Greenland and the Gobi desert. IPSL-CM5A-
LR and MR are virtually insensitive to reanalysis choice (compare Figure 5 with the "figs-refjra55/maps/rank" folder in the
supplementary material).

### 4.4   Model contributions from China and Japan

The *Beijing Climate Center Climate System Model* (BCC-CSM) comprises the BCC-AGCM3 AOGCM, originating from
CAM3 and developed independently thereafter (Wu et al., 2008), the completely independent land-surface model BCC-AVIM
developed by the *Chinese Academy of Science* (Jinjun, 1995) and GFDL's MOM and Sea Ice Simulator (SIS). For BCC-CSM2-
MR, the standard coupled model version used in CMIP6 (Wu et al., 2019), the latest updates of the in-house models are used in
conjunction with the CMIP5 versions of MOM and SIS (v4 and 2 respectively). The MAE and ranking patterns of BCC-CSM2-
MR are quite similar to those obtained from CCSM2 (compare Figure 6 and 3), which is likely due to the common origin of
their AGCMs, meaning that BCC-CSM2-MR is likewise found to perform comparatively poor in most regions of the NH. The
similarity between both model families is astonishing since they only share the *origin* of their atmospheric component but rely
on different land-surface, ocean and sea-ice models. This in turn means that the latter two components do not noticeably affect





the simulated atmospheric circulation as defined here, which is in line with the large differences found for the French models
in spite of using the same ocean model (see Section 4.3).

The *Nanjing University of Information Science and Technology Earth System Model version 3* (NESM3) is a new CMIP participant and is entirely built upon component models from other institutions (Cao et al., 2018). Namely, the AGCM, land-surface model, coupling software and even the atmospheric resolution are adopted from MPI-ESM1.2-LR (see Section 4.5) whereas NEMO3.4 and CICE4.1 are taken from IPSL and NCAR respectively (Cao et al., 2018). As a results, the error and
ranking patterns for NESM3 are similar to those obtained for MPI-ESM1.2-LR (compare Figure 6 with Figure 7). Exceptions are found over the central an western North Pacific, where NESM3 performs poorer than MPI-ESM1.2-LR, and also over the eastern North Pacific, where NESM3 performs better. The similarity to MPI-ESM1.2-LR again points to the fact that LWT frequency is determined by the AGCM rather than other component models.

The *Model for Interdisciplinary Research on Climate* (MIROC) relies on long-standing research efforts of the Japanese
research community led by the *Center for Climate System Research* (CCSR), the *National Institute for Environmental Studies* (NIES) and the *Japan Agency for Marine-Earth Science and Technology* (JAMESTEC). It comprises the *Frontier Research Center for Global Change* (FRCGC) AGCM and CCSR's *Ocean Component Model* (COCO), as well as an own land-surface (MATSIRO) and sea-ice model. MIROC5 and 6 are considered AOGCMs (Watanabe et al., 2010; Tatebe et al., 2019) whereas MIROC-ESM and MIROC-ESL2L are ESMs including atmospheric chemistry (Watanabe et al., 2011; Hajima et al., 2020).
Results indicate a systematic performance increase from MIROC5 to MIROC6 in the presence of large performance differences from one region to another (see Figure 6). Both models perform very well for the Mediterranean, northwestern North America and East Asia but do a poor job in northeastern North America and northern Eurasia. MIROC6 outperforms MIROC5 in the entire North Pacific basin including Japan, Corea and western North America and is also better in the central North Atlantic. On the contrary, MIROC5 only does better in southwestern North America. The performance of the two ESM versions is
considerably lower, both ranking unfavourably within the ensemble considered here.

Unarguably one of the most comprehensive representations of the Earth System is provided by *Japan's Meteorological Research Institute* (MRI). Already in the CMIP5 version of their ESM (MRI-ESM1), an atmospheric (photo)chemistry model coupled, an aerosol model and even a simple ice-sheet scheme was included in addition to the land and ocean carbon-cycle schemes necessary to form an ESM. The coupling applied in the MRI models is also more comprehensive than in most other
models (Yukimoto et al., 2011). Noteworthy, each model component and also the coupler have been originally developed by MRI. The comparatively high model resolution traditionally applied in this model family was further improved from MRI-ESM1 to MRI-ESM2 (Yukimoto et al., 2019) by adding vertical layers, particularly in the atmosphere (see Table 1). Especially when taking into account their complexity, both MRI models perform well in comparison with other models, except for the central to western Pacific basin including western North America, the subtropical North Atlantic to the west of the Strait of
Gibraltar, and the regions around Greenland and the Caspian Sea. It is in these "weak" regions where the largest performance gains are obtained from MRI-ESM1 to MRI-ESM2. As a results, in a zonal belt extending from approximately 50°N to 75°N, MRI-ESM2 is one of the best performing models considered here.





In the southwestern U.S. and around Greenland, the MRI models surprisingly agree closer with ERA-Interim than with the JRA-55 reanalysis also developed at JMA (compare Figure 6 with the "figs-refjra55/maps/rank" folder in the supplementary

material). For the MIROC family, a heterogeneous picture is obtained. While MIROC5 and MIROC-ESM clearly thrive towards ERA-Interim and JRA-55, respectively, MIROC6 is closer to JRA-55 in the southwestern U.S. and closer to ERA-Interim in the Gobi desert and around Greenland. The results for MIROC-ES2L, and also for BCC-CSM2-MR and NESM3, are virtually insensitive to reanalysis uncertainty.

### 4.5    Model contributions from Germany and Italy

The *Max-Planck Institute Earth System Model* (MPI-ESM) is another example for the synthesis of long-standing research efforts from many research institutes around the world, coordinated by the Max-Planck Institute for Meteorology (MPI-M) in Germany, with all component models developed independently. It comprises the ECHAM, JSBACH, MPIOM and HAMMOCC models representing the atmosphere, land-surface and terrestrial biosphere processes, ocean and sea-ice dynamics as well as ocean biogeochemistry, respectively, which are coupled with the OASIS software (Giorgetta et al., 2013; Jungclaus et al., 2013;

Mauritsen et al., 2019). Since atmospheric chemistry and aerosols are missing in all model versions except MPI-ESM1.2-HAM, these are generally less complex than e.g. the MRI model configurations mentioned above, but nevertheless include the carbon-cycle and are thus considered ESMs. The "working horse" used for generating large ensembles and long control runs is the "LR" version applied in both MPI-ESM-LR and MPI-ESM1.2-LR (for CMIP5 and 6, respectively). In this configuration, ECHAM (version 6 and 6.3) is run with a horizontal resolution of 1.9° (T63) and 47 layers in the vertical, and MPIOM with a

1.5° resolution near the equator and 40 levels in the vertical. In MPI-ESM-MR, the number of vertical layers in the atmosphere is doubled and the horizontal resolution in the ocean augmented to 0.4° near the equator. In MPI-ESM1.2, several atmospheric parametrization schemes have been improved and/or corrected, including radiation, aerosol, clouds, convection and turbulence, and the land-surface and ocean biogeochemistry processes have been made more comprehensive. In MPI-ESM1.2-LR the distribution of vegetation and landuse changes are simulated online, whereas they are prescribed in MPI-ESM1.2-HR, meaning

that this high-resolution version is on the limit to be considered an ESM. Since the carbon-cycle has not been run to equilibrium either, MPI-ESM1.2-HR is considered unstable by its development team Mauritsen et al. (2019). For MPI-ESM1.2-HAM, an aerosols and sulphur chemistry module, developed by a consortium led by the *Leibniz Institute for Tropospheric Research*, are coupled with ECAM6.3 in a configuration that otherwise is identical to MPI-ESM1.2-LR (Tegen et al., 2019). Similarly, *Alfred Wegener Institute's* AWI-ESM-1.1-LR makes use of their in-house ocean and sea-ice model FESOM but otherwise is identical

to MPI-ESM1.2-LR (Semmler et al., 2020).

Results show that the vertical resolution increase in the atmosphere undertaken from MPI-ESM-LR to MR (the CMIP5 versions) sharpens the regional performance differences rather than contributing to an improvement (see Figure 7). When switching from MPI-ESM-LR to MPI-ESM1.2-LR, i.e. from CMIP5 to 6 with constant resolution, the performance on the one hand increases for Europe but on the other decreases in most of the remaining regions. Notably, MPI-ESM-LR's good to very

good performance in a zonal belt ranging from the eastern subtropical North Pacific to the eastern subtropical Atlantic is lost in MPI-ESM1.2-LR. MPI-ESM1.2-HAM and AWI-ESM-1.1-LR worsen this picture and, even more so than MPI-ESM-MR,





are characterized by large regional performance differences and particularly unfavourable results over almost the entire North Pacific basin. However, *systematic* performance gains are obtained by MPI-ESM1.2-HR, indicating that a horizontal rather than vertical resolution increase in the atmosphere conducts to a better performance in this model family (recall that the sole *vertical*
resolution increase from MPI-ESM-LR to MPI-ESM-MR worsens the results). In the "HR" configuration, MPI-ESM1.2 is one of the best performing ESMs within the ensemble considered here.

The Centro Euro-Mediterraneo per i Cambiamenti Climatici (CMCC) models are mainly built upon component models from MPI, NCAR and IPSL. For CMCC-CM, ECHAM5 is used in conjunction with SILVA, a land-vegetation model developed in Italy (Fogli et al., 2009), and OPA8.2 (note that later OPA versions were integrated into the NEMO framework) plus LIM for
ocean and sea-ice dynamics, respectively. The very high horizontal resolution in atmosphere (T159) is achieved at the expense of a low horizontal resolution in the ocean and comparatively few vertical layers in both realms. Note that the carbon-cycle is not represented in this model version (Scoccimarro et al., 2011). For the core model contributing to CMIP6 (CMCC-CM2), CMCC substituted all of the aforementioned components except the OGCM by the those available from CESM1 (Hurrell et al., 2013). For the model version considered here (CMCC-CM2-SR5), CAM5.3 is run in conjunction with CLM4.5, taking
into account terrestrial carbon cycle processes. For ocean and sea-ice dynamics, NEMO3.6 (i.e. OPA's successor) and CICE are applied (Cherchi et al., 2019). The coupler changed from OASISv3 to CPLv7 (Valcke, 2006; Craig et al., 2012). Since ocean biogeochemistry processes are missing, none of the two model versions considered here reach ESM status. Due to the completely distinct model setups, the error and ranking patterns substantially change from CMIP5 to 6 for this model family. While CMCC-CM performs relatively weak in northern Canada, Scandinavia and northwestern Russia, CMCC-CM2-SR5
does so in the North Atlantic, particularly to the west of the Strait of Gibraltar. In the remaining regions, very good ranks are obtained by both models. Notably, CMCC-CM2-SR5 is one of the few models performing well in the central Asian high mountain ranges and also in the Rocky Mountains (except in Alaska). In most of the remaining regions it is likewise one of the best models considered here. Note that this model, due to identical model components for all realms except the ocean, is a good estimator for the performance of CESM1, which unfortunately cannot be assessed here due to data availability issues.
In the southwestern U.S. and around Greenland, the MPI models including AWI-ESM-1-1-LR and CMCC-CM consistently thrive towards JRA-55. On the contrary, CMCC-CM2-SR5 is in closer agreement with ERA-Interim, reflecting the profound change in the model components from CMIP5 to 6 (compare Figure 7 with the "figs-refjra55/maps/rank" folder in the supplementary material).

## 4.6 The joint European and Norwegian model contributions

The EC-Earth consortium is a large collaborative effort made by research institutions from several European countries. Following the idea of seamless prediction (Palmer et al., 2008), the atmospheric component used in the EC-Earth model is based on ECMWF's Integrated Forecasting System (IFS) complemented by the HTESSEL land-surface model and a new parametrization scheme for convection, NEMO for the ocean, LIM for sea-ice and OASIS being the coupler (Hazeleger et al., 2010, 2011). Starting from this basic AOGCM configuration, additional components of the extended climate system can be optionally added
to bring the model to a comprehensive ESM. However, most of the configurations used to produce historical experiments for





CMIP5 and 6 are classical AOGCMs and none of the versions analysed here reaches ESM status since ocean biogeochemistry is missing so far (see Table 1). A model version incorporating an interactive carbon cycle (EC-Earth3-CC) was not available at the time of submission and will be included in the final version of the manuscript. The version used in CMIP5 (EC-Earth2.3 or simply EC-Earth) already comprises a fine resolution in the atmosphere and ocean (except for the relatively few vertical

layers in the ocean) and this configuration was adopted or even improved for what is named "low resolution" in CMIP6 (EC-Earth3-LR, EC-Earth3-Veg-LR). For the other configurations used in CMIP6 (EC-Earth3 and EC-Earth3-Veg), the atmospheric resolution is further refined in the horizontal and vertical (Döscher et al., 2021). Results reveal an already very good performance for EC-Earth2.3 in all regions except the North Pacific and subtropical central Atlantic (see Figure 8) which, for the North Atlantic - European sector, is in line with the results from Perez et al. (2014) and Otero et al. (2017). EC-Earth3 performs

even better, and does so irrespective of the applied model complexity (vegetation dynamics are optionally added in EC-Earth3-Veg) or model resolution. These CMIP6 model versions, all run at the Irish Centre for High-End Computing (ICHEC), rank very well in almost any region of the the world, including the central Asian high mountain areas. As a results, if neither model complexity nor reanalysis uncertainty was an argument, then this model family would be claimed "the best one" in the context of the present study. Note that the very favourable results for EC-Earth3 hold for *any* of the 20 historical runs available from

the ESGF, with only slight variations in the error pattern and magnitude from one member of the initial conditions ensemble to another, meaning that internal climate variability plays a minor role here (see Section 4.8).

The Norwegian Earth System Model NorESM shares substantial parts of its source code with the NCAR model family (particularly with CCSM and CESM2). NorESM1-M, the standard model version used in CMIP5 (Bentsen et al., 2013), comprises the CAM4-Oslo AOGCM —derived from CAM4 and complemented with the Kirkevag et al. (2008) aerosol module—, CLM4

for land-surface processes, CICE4 for sea-ice dynamics and an ocean model based on the Miami Isopycnic Coordinate Ocean Model (MICOM) originally developed by NASA/GISS (Bleck and Smith, 1990). CPL7 is used as coupler. Biogeochemical processes can be included, but are not considered in the model version assessed here. From NorESM1 to NorESM2, the model components from CCSM were updated to CESM2.1 (Danabasoglu et al., 2020) whilst keeping the Norwegian aerosol module and modifying a number of parametrization schemes in CAM6-Nor w.r.t. to CAM6 (Seland et al., 2020). Through the cou-

pling of an updated MICOM version with the ocean biogeochemistry model HAMOCC, combined with the use of the CLM5 land-surface model, both oceanic and terrestrial biogeochemical processes are represented in NorESM2. Since the Community Ice Sheet Model (CISM) is used in addition (Lipscomb et al., 2019), NorESM2 pertains to the group of the most sophisticated ESMs considered here (together with MRI-ESM1 MRI-ESM2 and HadGEM2-ES). The coupler has been updated from CPL7 to CIME, which is also used in CESM2. In the present study, the basic configuration NorESM2-LM is evaluated together with

NorESM2-MM, the latter being integrated with a much finer horizontal resolution in the atmosphere (see Table 1). Otherwise, the two experiments are identical. The corresponding maps in Figure 8 reveal a low model performance for NorESM1-M with an error magnitude and spatial pattern similar to CCSM4. When switching to NorESM2-LM, i.e. to updated and extended component models and an almost identical resolution in the atmosphere and ocean, notable performance gains are obtained in most regions of the NH, except in a zonal band extending from Newfoundland to the Urals which, further to the East, re-emerges





over the Baikal region. In the higher-resolution version NorESM2-MM these errors are further reduced to a large degree, with the overall effect of obtaining one of the best models considered here, particularly when its complexity is taken into account.

If these two model families are evaluated against JRA-55 instead of ERA-Interim, the ranks for the EC-Earth model family consistently worsen by up to 20 integers in the southwestern U.S. and around the southern tip of Greenland, but remain roughly constant in the Gobi desert (compare Figure 8 with the "figs-refjra55/maps/rank" folder in the supplementary material). This worsening brings the EC-Earth family to a closer agreement with the HadGEM models. Consequently, when evaluated against JRA-55, HadGEM3-GC31-MM links up with EC-Earth3 in what is considered the "best model" if model complexity was not argument (see also "figs-refjra55/as-figure-10-but-wrt-jra55.pdf" in the supplementary material). For the NorESM family, different reanalysis affinities are obtained for the 3 regions. While NorESM1 is closer to JRA-55 in all of them, NorESM2-LM is closer to ERA-Interim in the southwestern U.S. but closer to JRA-55 in the Gobi. NorESM2-MM is generally less sensitive to reanalysis uncertainty, with some affinity to ERA-Interim in the southwestern U.S.

### 4.7 Model contributions from Russia and South Korea

The *Institute of Numerical Mathematics, Russian Academy of Sciences* model INM-CM4 is a classical AOGCM comprising an atmosphere, land-surface, ocean and sea-ice model, all developed by scientists working Russia (Volodin et al., 2010). INM-CM4 contributed to CMIP5 and an updated version (INM-CM4-8) is currently participating in CMIP6, but the 6-hourly SLP data is not available for this version so that it had to excluded here. The resolution setup of INM-CM4 is comparable to other CMIP5 models, except for the very few vertical layers used in the atmosphere (see Table 1). As shown by Figure 9, INM-CM4 performs well to very well in the eastern North Atlantic, northern Europe and the Gulf of Alaska, regularly over northern China and Corea and poorly over the remaining regions of the NH. It is thus marked by large performance differences from one region to another.

The *Seoul National University Atmosphere Model version 0 with a Unified Convection Scheme* (SAM0-UNICON) contributes for the first time in CMIP6 (Park et al., 2019). Its component models are identical to CESM1 in its AOGCM configuration including aerosols (Hurrell et al., 2013), with the special feature of using a large number of alternative parametrization schemes involving convection, stratiform clouds, aerosols, radiation, surface fluxes and planetary boundary layer dynamics (Park et al., 2019). In spite of its conceptional similarity to CMCC-CM2-SR5 and NorESM2, the error *pattern* is different in SAM0-UNICON (compare Figure 9 with Figures 8 and 7), which might be due to the ocean model taken from CESM1 (POP is used instead of NEMO or MICOM, see Table 1), or precisely due to the effects of the particular parametrization schemes mentioned above. Although the error *magnitude* of SAM0-UNICON is similar to CMCC-CM-SR5, SAM0-UNICON exhibits weaker regional performance differences, making it the more balanced model out of the two. In most regions of the NH, SAM0-UNICON yields better results than NorESM2-LM but is outperformed by NorESM2-MM.

While INM-CM4 compares better with JRA-55 in the 3 regions sensitive to reanalysis uncertainty, SAM0-UNICON is in closer agreement with ERA-Interim, there (compare Figure 9 with the "figs-refjra55/maps/rank" folder in the supplementary material).





## 4.8 Summary boxplot and role of internal model variability

For each model version listed in Table 1, the spatial distribution of the pointwise MAE values can also be represented with a
boxplot instead of a map, which allows for an overarching performance comparison visible at a glance (see Figure 10 for the
evaluation against ERA-Interim). Here, the standard configuration of the boxplot is applied. For a given sample of MAE values
corresponding to a specific model, the box refers to the interquartile range (IQR) of that sample and the horizontal bar to the
median. Whiskers are drawn at the 75th percentile + 1.5 × IQR and at the 25th percentile - 1.5 × IQR. All values outside
this range are considered outliers (indicated by dots). Additional boxplots are provided for the joint MAE samples of 1) all
CMIP5 model versions, 2) all CMIP6 model versions, 3) all model versions considered ESMs (ESM) and 4) all other model
versions (AOGCM). In these 4 cases the outliers are not plotted for the sake of simplicity. The acronyms of the *coupled* model
configurations, as well as their participation in either CMIP5 or 6 (indicated by the final integer), are shown below the x-axis.
Above the x-axis, the names of the coupled models' *atmospheric* components are also shown since some of them are shared
by various research institutions (see also Table 1).

Results indicate a performance gain for most model families when switching from CMIP5 to 6 (available model pairs are
located next to each other in Figure 10). The largest improvements are obtained for those models performing relatively poorly
in CMIP5. Namely, NorESM2-LM and NorESM2-MM improve upon NorESM1-M (rose), MIROC6 improves upon MIROC5
(blue-green) and IPSL-CM6A-LR upon IPSL-CM5A-LR and IPSL-CM5A-MR (grey). GISS-E2-R-5 improves upon GISS-
E2-H and GISS-E2-R (green) in terms of *median* performance, but suffers slightly larger spatial performance differences as
indicated by the IQR. The MPI (neon green), CMCC (cyan), GFDL (magenta) and MRI (brown) models already performed
well in CMIP5 and further improve in CMIP6. Among the MPI models, however, an advantage over the two CMIP5 versions is
only obtained when considering the *high-resolution* CMIP6 version (compare MPI-ESM1.2-HR with MPI-ESM-LR and MPI-
ESM-MR). Contrary to the remaining models, the performance of the CNRM (red) models does *not* improve from CMIP5 to 6,
which may be due to the fact that the CMIP5 version (CNRM-CM5) already performed very well. Remarkably, CNRM's high-
resolution CMIP6 version (CNRM-CM6-1-HR) is the worst performing one within this model family. Similary, the ACCESS
models (blue) do not improve either if ACCESS1.0 instead of ACCESS1.3 is taken as reference CMIP5 model. The CMCC,
HadGEM, and particularly the EC-Earth model families perform overly best and all three exhibit a performance gain from
CMIP5 to 6. However, none of the EC-Earth or CMCC model versions is an ESM and neither is HadGEM3-GC31-MM, the
latest Hadley Centre model version considered here. So if model complexity matters and only ESMs are taken into account,
then NorESM2-MM is the best choice, followed by MRI-ESM2, GFDL-CM4, MPI-ESM1.2-HR and ACCESS-ESM1.5, as
well as the HadGEM2-ES model already used in CMIP5. Given its status as fully comprehensive ESMs, the virtual lack of
outliers is another remarkable advantage of NorESM2-MM. MRI-ESM2 and GDFL-CM4 are also relatively robust to outliers,
but less so than NorESM2-MM. The fewest number of outliers among all models is obtained for EC-Earth, irrespective of the
model version.

The model evaluation against JRA-55 reveals similar results (see "figs-refjra55/as-figure-10-but-wrt-jra55.pdf" in the sup-
plementary material), indicating that uncertain reanalysis data in the 3 relevant regions detected above do do not substantially



affect the hemispheric-wide statistics. What is noteworthy, however, is the slight but nevertheless visible performance loss for the EC-Earth model family, bringing EC-Earth3 approximately to the performance level of HadGEM3-GC31-MM. If evaluated against JRA-55, all EC-Earth model versions also comprise more outlier results.

Table 2 provides the rank correlation coefficients between the median MAE w.r.t. to ERA-Interim for each model (corresponding to the horizontal bars within the boxes in Figure 10) and various resolution parameters for the atmosphere and/or ocean component models. Correlations are calculated separately for the zonal, meridional and vertical resolution represented by the number of grid-boxes in the corresponding direction (due to the presence of reduced Gaussian grids, longitudinal grid-boxes *at the equator* are considered), as well as for the 2D (lon × lat) and 3D (lon × lat × layers) meshes, respectively. This is

done separately for the atmosphere and ocean. Due the presence of an unstructured grid in one ocean model, the breakdown in zonal and meridional resolution cannot be made in this realm. In a final step, the number of grid-boxes of the 3D meshes from both atmosphere *and* ocean are added to obtain the size of combined atmosphere-ocean mesh.

As can be seen from Table 2, average model performance is closer related to the *horizontal* than to the vertical resolution applied in the atmosphere. Associations with the ocean resolution are weaker, as expected, but nevertheless significant for both

the horizontal and vertical resolution. This is somewhat unexpected, particularly when taking into account that the corresponding link with the vertical resolution in the *atmosphere* is spurious. Since the resolution increase for most models has gone hand in hand with improvements in the internal parameters (parametrization, model physics, bugs) it is difficult to say which of these two effects is more influential on model performance. However, most of the models undergoing a version change *without* resolution increase do not experience a clear performance gain either. This is observed for the 3 ACCESS versions using

the same AGCM (i.e. GA in 1.3, CM2 and ESM1-5) and also for the 3 model versions from GISS, all comprising the same horizontal resolution in the atmosphere within their respective model family. Likewise, CNRM-CM6-1 and MPI-ESM1-2-LR even perform slightly worse than their predecessors (CNRM-CM5 and MPI-ESM-LR), meaning that the update is counterproductive for their performance (see Figure 10). This points to the fact that resolution is likely more influential on performance than model updates as long as the latter are not too substantial.

In comparison with the *inter*-model variability discussed above, the *internal* model variability (or "intra-model variability") is much smaller and only marginally affects the results, which for all runs of a given model version are in close agreement even for the outliers (see Figure 11). Albeit the use of alternative model runs might lead to slight shifts in the ranking order at the grid-box scale, which is why a "best model per grid-box map" is intentionally not provided here, a "good" rank would not change into an "average" or even "bad" one. However, while internal model variability does only play a minor role in the

context of the present study, some specific models indeed seem to be more sensitive to initial conditions uncertainty (which is where ensemble spread stems from in the experiments considered here) then others, with NorESM2-LM (the lower resolution version only) and NESM3 seemingly being less stable in this sense. Remarkably, MPI-ESM1.2-HR is found to be stable in spite of the fact that it is considered a more "unstable" configuration by its development team because the carbon cycle had not been run been to equilibrium for this version of MPI-ESM1.2 (Mauritsen et al., 2019). It is also good news that HadGEM2-ES,

known to perform well for *r1i1p1* and consequently used as baseline for many downscaling applications and impact studies



(Perez et al., 2014; Gutiérrez et al., 2013; San-Martín et al., 2016), performs nearly identical for *r2i1p1*. Finally, the large performance increase from IPSL-CM5A-LR to IPSL-CM6A-LR is likewise robust to the effects of internal model variability.

## 5 Specific model performance for each Lamb weather type

In Figures 12 to 14, the simulated, hemispheric-wide frequency pattern for a given model and LWT is compared with the
respective quasi-observed frequency pattern obtained from ERA-Interim by using a normalized Taylor diagram (Taylor, 2001). The first thing to note here is that, for most LWTs, the models tend to cluster in a region that would be generally considered a good result. Except for some outlier models and individual LWTs, the pattern correlation lies in between 0,6 and 0.9, the standard deviation ratio is not too far from unity (= best result) and the centred normalized RMSE ranges between 0.25 and 0.75 × the standard deviation of the observed frequency pattern.

It is also becomes evident that all members of the EC-Earth model family yield best results for *any* LWT (observe the proximity of the yellow cluster to the perfect score indicated by the black half cycle). Recall, however, that no EC-Earth version actually fullfils the criterion of an ESM since ocean biogeochemistry is not considered. Within the group of ESMs, NorESM2-MM (the rose triangle pointing to the left) performs best and actually lies in close proximity to the EC-Earth Cluster for most LWTs. The Hadley Centre and ACCESS models (filled with orange and dark blue) form another cluster that generally
performs very well for most LWTs. However, the spatial standard deviation of the 3 *eastern* LWTs (cyclonic, anticyclonic and directional) is overestimated by these "Commonwealth" models (the Commonwealth is here referred to for illustrative purposes and does not reflect any political opinion), which is indicated by a standard deviation ratio ≈1.25, while values close to unity or below are obtained for the remaining models. It is also worth mentioning that not only ACCESS1.0 but also the other, more independently developed ACCESS versions pertain to the Commonwealth cluster, which indicates the common
origin of their atmospheric component (the Met Office Hadley Centre) even at the level of detail of specific weather types. For all other models, the LWT-specific results do not largely deviate from the overall MAE results shown in Section 4, meaning that overall performance is generally also a good indicator of LWT-specific performance. As an example, MIROC-ESM (the blue-green cross), IPSL-CM5A-LR and IPSL-CM5A-LR (the grey cross and grey plus) are located in the "weak" area of the Taylor diagram for *each* of the 27 LWTs, which is in line with the likewise weak *overall* performance obtained for these models
in Section 4.

The corresponding results for the model evaluation against JRA-55 are generally in close agreement with those mentioned above, except for the EC-Earth model family performing slightly less favourable (see "figs-refjra55/taylor" folder in the supplementary material to this article).

## 6 Summary, discussion and conclusions

In the present study, 46 coupled general circulation model versions contributing historical experiments to CMIP5 and 6 have been evaluated in terms of their capability to reproduce the observed frequency of the 27 atmospheric circulation types orig-





inally proposed by Lamb (1972), as represented by the ERA-Interim or JRA-55 reanalyses. The outcome is an objective, regional-scale ranking catalogue that is expected to be of interest for the model development teams themselves, and also for the downscaling and regional climate change community asking for model selection criteria. In this context, the present study
is a direct response to the claim for a circulation-based model performance assessment made by Maraun et al. (2017).

On average, the model versions used in CMIP6 perform better than their CMIP5 predecessors and the more complex ESMs are outperformed by the simpler AOGCMs. Among a number of tested resolution parameters, the closest statistical relationship with model performance is obtained for the horizontal resolution in the atmosphere, which is in line with Cannon (2020), with equal contributions from the latitudinal and longitudinal grid distance and no significant relationship for the number of vertical
layers. The corresponding links with the ocean resolution are weak but nevertheless significant, even for the number of vertical layers used in this realm. Improving the internal model parameters (physics and parametrization schemes) and/or adding more vertical layers to the atmosphere seems to have little effect for most model families if the horizontal resolution is not refined in addition. This is the case for ACCESS-CM2 w.r.t. ACCESS1.3, CNRM-CM6-1 w.r.t. CNRM-CM5, GISS-E2-1-G w.r.t. GISS-ES-R and MPI-ESM1.2-LR w.r.t. MPI-ESM-LR.

For a subgroup of 12 out of 46 models, the impact of internal model variability on the performance was assessed with 70 additional historical integrations, each one initialized from a unique starting date of the corresponding pre-industrial control run. The thereby created initial conditions uncertainty has little effect on the overall results. Albeit the point-wise ranking order might change by a few integers when alternative runs are evaluated, which is why a "best model" map is intentionally not provided here, a well performing model would not even change to an "intermediate" one or vice versa if another ensemble
member was put to the test. A similarly small effect was found for changing the reference reanalysis from ERA-Interim to JRA-55, except in the following 3 problematic regions, where this change can largely affect the models' ranking order: the southwestern United States, the Gobi desert, and Greenland plus the surrounding seas.

This study also shows that the models' complexity, here defined as the number of realms simulated online, should be taken into account for a correct interpretation of the results. Namely, comprehensive ESMs such as HadGEM2-ES, MRI-ESM1,
MRI-ESM2 and NorESM2 are by construction more sensitive to model uncertainties than traditional AOGCM configurations. Hence, while the distinct EC-Earth versions considered here are consistently performing "best", none of them reaches the complexity of an ESM. This specific conclusion will be re-evaluated by inclusion of EC-Earth3-CC during the review phase, which was not available when this manuscript version was submitted. If only ESMs are considered, NorESM2-MM and MRI-ESM2 play a particular role because they are the most complex models and at the same time perform comparatively well, a
finding that also holds for the older, well tested, HadGEM2-ES. GFDL-CM4 and MPI-ESM1.2-HR perform similarly well but are less complex than NorESM2-MM, MRI-ESM2 and HadGEM2-ES.

Since ESMs are in principle preferable to AOGCMs, a discussion about how model complexity should influence the choice of driving GCMs in regional climate studies is needed. A separate ranking of the models pertaining to each group would be a simple solution (see "figs-refinterim-aogcm" and "figs-refinterim-esm" folders in the supplementary material to this article).
Once the user has decided on whether to use AOGCMs or ESMs, he/she can then select the most favourable model(s) from one of the two groups.



*Code and data availability.* The netCDF files containing the Lamb Weather Type catalogues computed for this study have been permanently archived at https://doi.org/10.5281/zenodo.4452081. The underlying Python code was stored at https://doi.org/10.5281/zenodo.4555368.

*Author contributions.* All working steps were accomplished by SB.

*Competing interests.* The author declares no competing interests.

*Acknowledgements.* The author would like to thank the *Agencia para la Modernización Tecnológica de Galicia* (AMTEGA) and the *Centro de Supercomputación de Galicia* (CESGA) for providing the necessary computational resources.





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

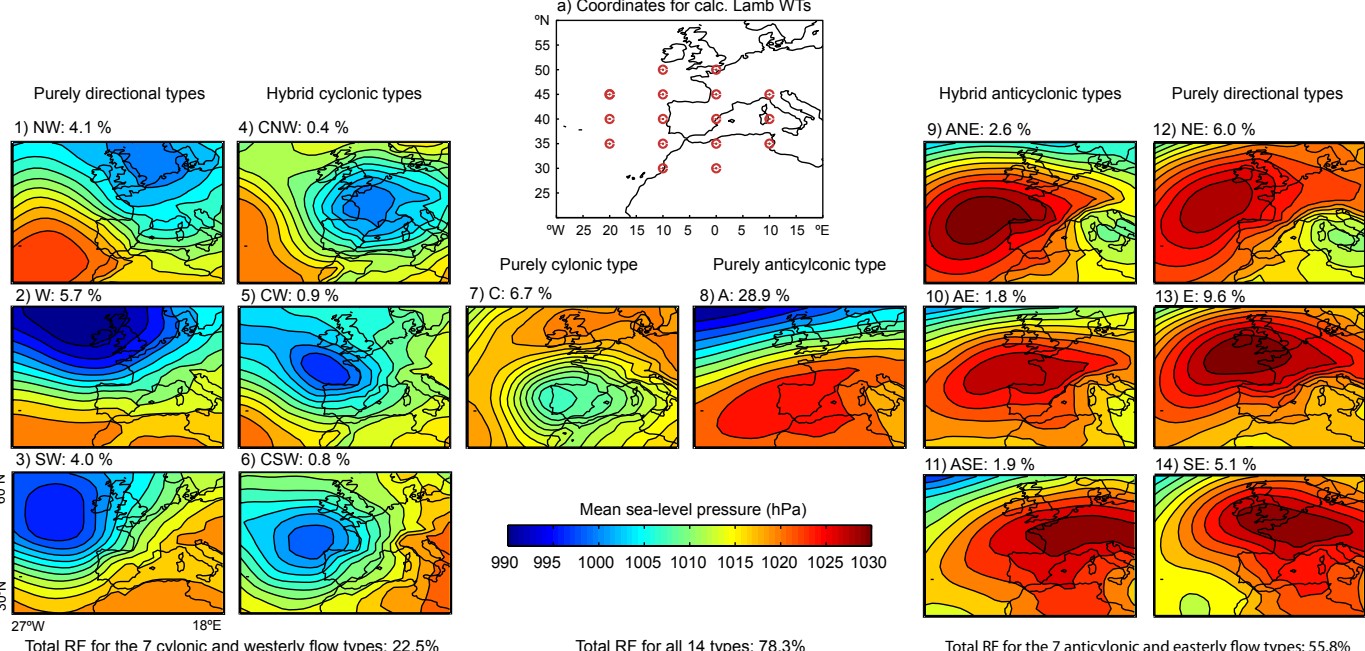

**Figure 1.** Illustrative example for the usage of the Lamb weather types approach over the central Iberian Peninsula. Shown is the coordinate system configured for this region and a subset of 14 types as well as their relative occurrence frequencies. Note that in the present study, all 27 types originally defined in Lamb (1972) are being used. The figure is taken from Brands et al. (2014), courtasy to John Wiley and Sons.





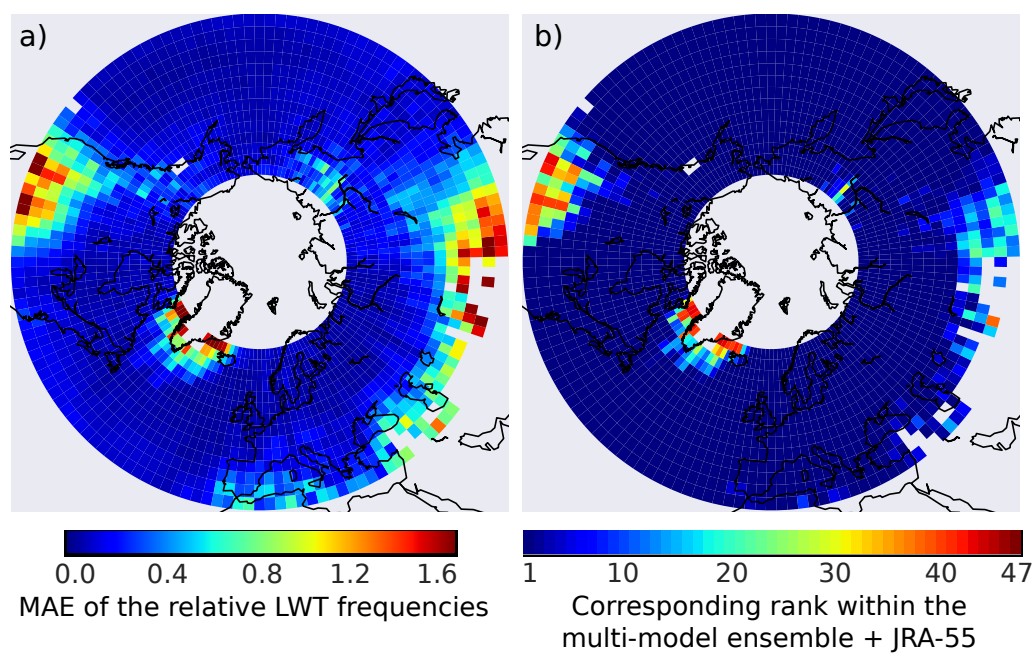

**Figure 2.** Mean Absolute Error of the relative Lamb weather type frequencies from JRA-55 w.r.t. to ERA-Interim (a), as well as the respective rank within the multi-model ensemble plus JRA-55 (b). The lower the rank, the lower the MAE and the closer the agreement between JRA-55 and ERA-Interim.







**Figure 3.** Mean Absolute Error of the relative Lamb weather type frequencies from the historical CMIP experiments w.r.t. to ERA-Interim (column a), as well as the respective rank within the 46 distinct model versions outlined in Table 1 (column b). The lower the rank, the lower the MAE and the better the model. Results are for the *Met Office Hadley Centre* and *ACCESS* model families. Model pairs from CMIP5 and 6 are plotted next to each other. Results are for the 1979-2005 period.





**Figure 4.** As Figure 2, but for the GFDL, GISS, CCCma and NCAR models.





**Figure 5.** As Figure 2, but for the CNRM and IPSL models







**Figure 6.** As Figure 2, but for the BCCR, NESM, MIROC and MRI models





**Figure 7.** As Figure 2, but for the MPI and CMCC models





**Figure 8.** As Figure 2, but for the EC-Earth and NorESM models





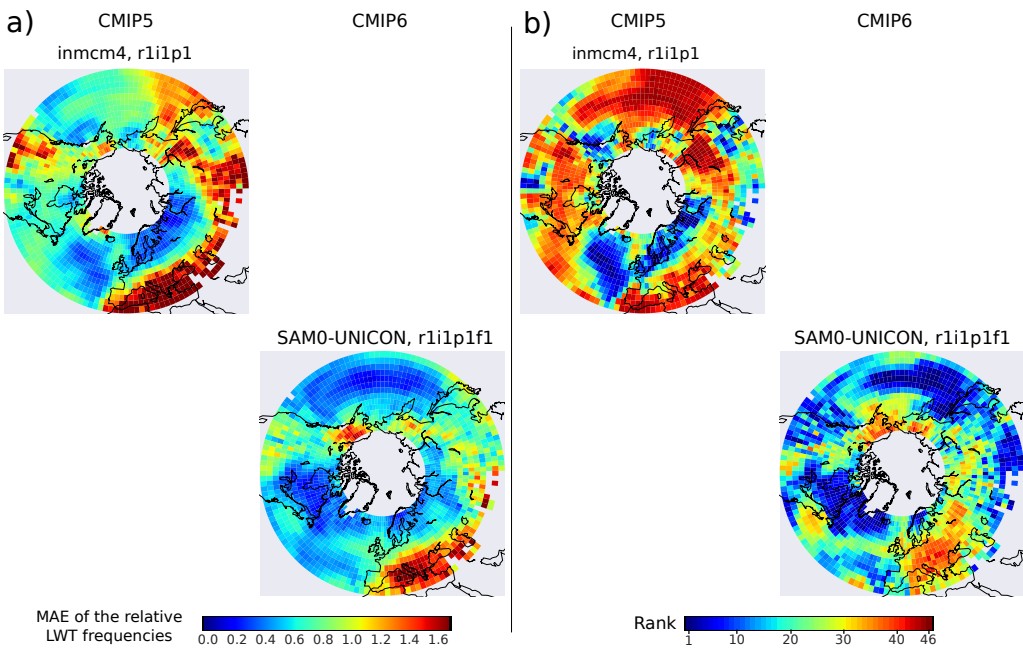

**Figure 9.** As Figure 2, but for INM-CM4 and SAM0-UNICON



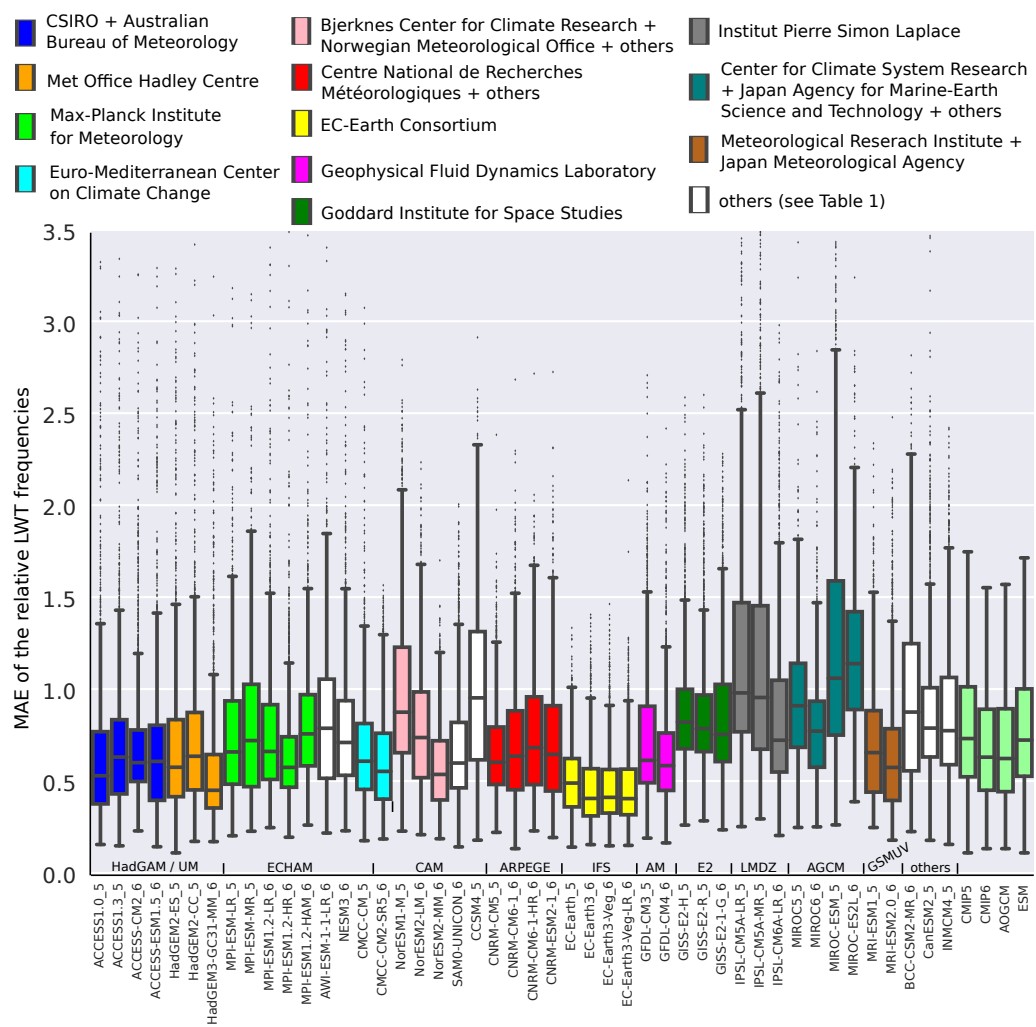

**Figure 10.** Summary model performance plot based on the MAE. For each model version listed in Table 1 the distribution of the pointwise MAE values is drawn with a boxplot instead of using a map (see text for details). Additional boxplots are provided for 1) all CMIP5 model versions, 2) all CMIP6 model versions 3) all Earth System Models (ESMs) and 4) the remaining models (AOGCM). Colours are assigned to the distinct coordinating research institutes, as indicated in the legend. The acronyms of the coupled models, as well as their participation in either CMIP5 or 6 (indicated by the final integer) are shown below the x-axis. Above the x-axis, the atmospheric component of each coupled model is shown in addition. Results are for the 1979-2005 period.



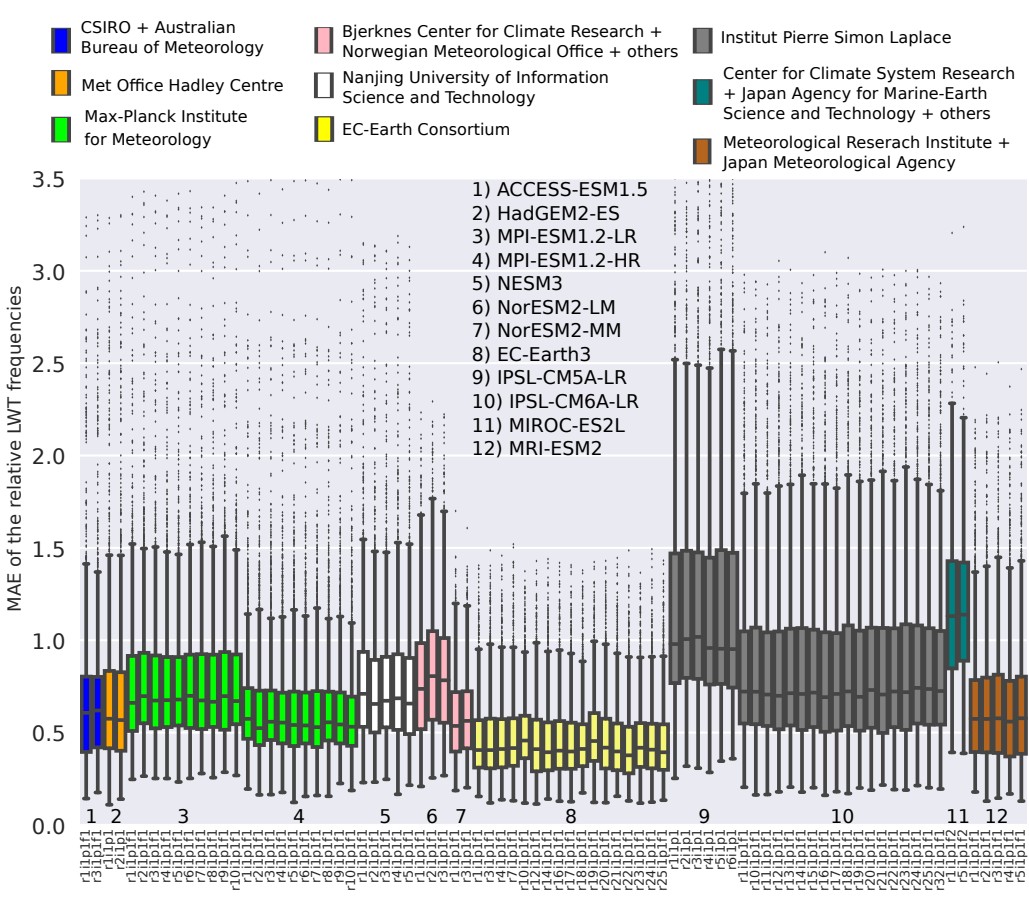

**Figure 11.** As Figure 10, but considering 70 additional runs for a subset of 12 distinct coupled models. All available runs per model are taken into account, except for IPSL-CM6A-LR for which the analyses were stopped after considering 17 additional ensemble members. Colours indicating the coordinating research institute are identical to Figure 9, except for the *Nanjing University of Information Science and Technology* painted white. Up to 2 ensembles per institute are shown and the acronyms of the individual coupled models are indicated by numbers. The exact run specifications are provided along the x-axis.





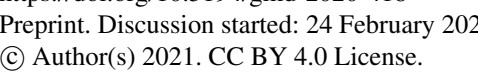

**Figure 12.** Normalized Taylor diagram for the simulated vs. quasi-observed (from ERA-Interim) hemispheric-wide frequency pattern of a given Lamb weather type. Each panel corresponds to a specific LWT and each of the 46 considered models can be identified by a specific marker and colour, as indicated in the legend. Models pertaining to the same coordinating institution have the same colour. Shown are the results for the 9 *anticylonic* Lamb weather types.

**Figure 13.** As Figure 11, but for the 8 *purely directional* Lamb weather types and the *unclassified* type.



**Figure 14.** As Figure 11, but for the 9 *cyclonic* Lamb weather types.

| Coupled Model | CMIP | Atmos. Model | Ocean Model | Historical Exp. | References | Complexity | Affinity |
|---|---|---|---|---|---|---|---|
| ACCESS1-0 | 5 | HadGAM2, 192 × 144, 38 lv | NOAA/GFDL MOM4p1, 360 × 300, 50 lv | r1i1p1 | Bi et al. (2013) | AOGCM+aero | mixed |
| ACCESS1-3 | 5 | UM7.3-approx. GA1, 192 × 144, 38 lv | NOAA/GFDL MOM4p1, 360 × 300, 50 lv | r1i1p1 | Bi et al. (2013) | AOGCM+aero | mixed |
| ACCESS-CM2 | 6 | UM10.6-GA7.1, 192 × 144, 85 lv | ACCESS-OM2 (GFDL-MOM5), 360 × 300, 50 lv | r1i1p1f1 | Bi et al. (2020) | AOGCM+aero | mixed |
| ACCESS-ESM1-5 | 6 | UM7.3-approx. GA1, 192 × 145, 38 lv | ACCESS-OM2 (GFDL-MOM5), 360 × 300, 50 lv | r1i1p1f1 + 1 | Ziehn et al. (2020) | ESM+aero | mixed |
| AWI-ESM-1-1-LR | 6 | ECHAM6.3.04p1, 192 × 96, 47 lv | FESOM 1.4, 126859 wet nodes (unstructed mesh), 46 lv | r1i1p1f1 | Semmler et al. (2020) | AOGCM | JRA-55 |
| BCC-CSM2-MR | 6 | BCC-AGCM3-MR, 320 × 160, 46 lv | GFDL-MOM4-MR, 360 × 232, 40 lv | r1i1p1f1 | Wu et al. (2019) | AOGCM | none |
| CanESM2 | 5 | CanAM4, 128 × 64, 35 lv | CanOM4, 256 × 192, 40 lv | r1i1p1 | Chylek et al. (2011) | ESM | JRA-55 |
| CCSM4 | 5 | CAM4, 288 × 192, 26 lv | POP2, 384 × 320, 60 lv | r6i1p1 | Gent et al. (2011) | AOGCM | Interim |
| CMCC-CM | 5 | ECHAM5, 480 × 240, T159, 31 lv | OPA8.2-ORCA2, 31 lv | r1i1p1 | Scoccimarro et al. (2011) | AOGCM | JRA-55 |
| CMCC-CM2-SR5 | 6 | CAM5.3, 288 × 192, 30 lv | NEMO3.6-ORCA1, 50 lv | r1i1p1f1 | Cherchi et al. (2019) | AOGCM+chem+tbgc | Interim |
| CNRM-CM5 | 5 | ARPEGE-Climat v5.2.1 256 × 128, 31 lv | NEMO3.2-ORCA1, 42 lv | r1i1p1 | Voldoire et al. (2013) | AOGCM | mixed |
| CNRM-CM6-1 | 6 | ARPEGE 6.3 256 × 128, 91 lv, T127 Gauss. red. 24572 grid points | NEMO3.6-ORCA1, 75 lv | r1i1p1f2 | Voldoire et al. (2019) | AOGCM+chem | mixed |
| CNRM-CM6-1-HR | 6 | ARPEGE 6.3, 720 × 360, 91 lv, T359 Gauss. red. 181724 grid points | NEMO3.6-ORCA025, 75 lv | r1i1p1f2 | Voldoire et al. (2019) | AOGCM+chem | mixed |
| CNRM-ESM2-1 | 6 | ARPEGE 6.3 720 × 360, T127 Gauss. red. 24572 grid points, 91 lv | NEMO3.6-ORCA1, 75 lv | r1i1p1f2 | Séférian et al. (2019) | ESM+chem+aero | mixed |
| EC-Earth-2.3 | 5 | IFS (cy31R1+modifications), 320 × 160, T159L62, 62 lv | NEMO+modifications-ORCA1, 42 lv | r12i1p1 | Hazeleger et al. (2011) | AOGCM | Interim |
| EC-Earth3 | 6 | IFS (IFS cy36r4), 512 × 256, T255L91 Gauss. red., 91 lv | NEMO3.6-ORCA1, 75 lv | r1i1p1f1 + 16 | Döscher et al. (2021) | AOGCM | Interim |
| EC-Earth3-Veg | 6 | IFS (IFS cy36r4), 512 × 256, T255L91 Gauss. red., 91 lv | NEMO3.6-ORCA1, 75 lv | r1i1p1f1 | Döscher et al. (2021) | AOGCM+tbgc | Interim |
| EC-Earth3-Veg-LR | 6 | IFS (IFS cy36r4), 320 × 160, T159L62 Gauss. red., 62 lv | NEMO3.6-ORCA1, 75 lv | r1i1p1f1 | Döscher et al. (2021) | AOGCM+tbgc | Interim |
| GFDL-CM3 | 5 | AM3p9, 144 × 90, C48L48, 48 lv | MOM4p1, 360 × 200, tripolar grid, 1/3° at the equator, 50 lv | r1i1p1 | Griffies et al. (2011) | AOGCM+chem+aero | mixed |
| GFDL-CM4 | 6 | GFDL-AM4.0.1, 360 × 180, Cubed-sphere, c96, 33 lv | GFDL-OM4p25 (GFDL-MOM6), 1440 × 1080, tripolar 0.25° grid, 75 lv | r1i1p1f1 | Held et al. (2019) | ESM+chem+aero | mixed |
| GISS-E2-H | 5 | GISS-E2, 144 × 90, 40 lv | Hycom, 1 × cos(lat) tripolar grid north of 58°, mercator below, 26 lv | r6i1p1 | Schmidt et al. (2014) | AOGCM+chem+aero | Interim |
| GISS-E2-R | 5 | GISS-E2, 144 × 90, 40 lv | Russel Ocean, 288 × 180, regular lat-lon, 32 lv | r1i1p1 | Schmidt et al. (2014) | AOGCM+chem+aero | Interim |
| GISS-E2-1-G | 6 | GISS-E2.1, 144 × 90, 40 lv | GISS Ocean, 288 × 180, regular lat-lon, 32 lv | r1i1p1f1 | Kelley et al. (2020) | AOGCM | none |
| HadGEM2-CC | 5 | HadGAM2, 192 × 145, N96L60, 60 lv | HadGOM2, 360 × 216, 40 lv | r1i1p1 | Collins et al. (2011) | ESM+chem | mixed |
| HadGEM2-ES | 5 | HadGAM2, 192 × 145, N96L38, 38 lv | HadGOM2, 360 × 216, 40 lv | r1i1p1 + 1 | Collins et al. (2011) | ESM+chem | mixed |
| HadGEM3-GC31-MM | 6 | UM10.6-GA7.1, 432 × 324, N216L85, 85 lv | NEMO-HadGEM3-GO6.0-eORCA025, 75 lv | r1i1p1f3 | Roberts et al. (2019) | AOGCM+aer+chem+tbgc | mixed |
| INMCM4 | 5 | INM-CM4 atmosphere model, 180 × 120, 21 lv | INM-CM4 ocean model, 360 × 360, 40 lv | r1i1p1 | Volodin et al. (2010) | AOGCM | JRA-55 |
| IPSL-CM5A-LR | 5 | LMDZ4v5, 96 × 95, 39 lv | NEMO3.2-ORCA2, 31 lv | r1i1p1 + 5 | Dufresne et al. (2013) | ESM | none |
| IPSL-CM5A-MR | 5 | LMDZ4v5, 144 × 143, 39 lv | NEMO3.2-ORCA2, 31 lv | r1i1p1 | Dufresne et al. (2013) | ESM | none |
| IPSL-CM6A-LR | 6 | LMDZ NPv6, 144 × 143, N96L79, 79 lv | NEMO-OPA-eORCA1.3, 75 lv | r1i1p1f1 + 17 | Boucher et al. (2020) | ESM | mixed |
| MIROC5 | 5 | MIROC-AGCM6, 256 × 128, T85L40, 40 lv | COCO4.5, 256 × 224, 50 lv | r1i1p1 | Watanabe et al. (2010) | AOGCM | Interim |
| MIROC-ESM | 5 | MIROC-AGCM 2010, 128 × 64, T42L80, 80 lv | COCO3.4, 256 × 192, 44 lv | r1i1p1 | Watanabe et al. (2011) | ESM+aero | JRA-55 |
| MIROC6 | 6 | CCSR AGCM, 256 × 128, T85L81, 81 lv | COCO4.9, 360 × 256, tripolar primarily 1° grid, 63 lv | r3i1p1f1 | Tatebe et al. (2019) | AOGCM+aero | mixed |
| MIROC-ES2L | 6 | CCSR AGCM, 128 × 64, T42L40, 40 lv | COCO4.9, 360 × 256, tripolar primarily 1° grid, 63 lv | r5i1p1f2 + 1 | Hajima et al. (2020) | ESM+aero | none |
| MPI-ESM-LR | 5 | ECHAM6, 192 × 96, T63L47, 47 lv | MPIOM, 256 × 220, bipolar grid with 1.5° near the equator, 40 lv | r1i1p1 | Giorgetta et al. (2013) | ESM | JRA-55 |
| MPI-ESM-MR | 5 | ECHAM6, 192 × 96, T63L95, 95 lv | MPIOM, 802 × 404, tripolar grid with 0.4° near the equator, 40 lv | r1i1p1 | Giorgetta et al. (2013) | ESM | JRA-55 |
| MPI-ESM1.2-LR | 6 | ECHAM6.3, 192 × 96, T63L95, 47 lv | MPIOM1.63, 360 × 256, bipolar grid with 1.5° near the equator, 40 lv | r1i1p1f1 + 9 | Mauritsen et al. (2019) | ESM | JRA-55 |
| MPI-ESM1.2-HR | 6 | ECHAM6.3, 384 × 192, T127L95, 95 lv | MPIOM1.63, 802 × 404 tripolar grid with 0.4° near the equator, 40 lv | r1i1p1f1 + 9 | Müller et al. (2018) | ESM | JRA-55 |
| MPI-ESM1.2-HAM | 6 | ECHAM6.3, 192 × 96, T63L95, 47 lv | MPIOM1.63, 256 × 220, bipolar grid with 1.5° near the equator, 40 lv | r1i1p1f1 | Mauritsen et al. (2019) | ESM+chem+aero | JRA-55 |
| MRI-ESM1 | 5 | GSMUV-1101 20oc, 320 × 160, TL159L48, 48 lv | MRICOM-3-0, 368 × 364, tripolar primarily 0.5 × 1.0° grid, 51 lv | r1i1p1 | Yukimoto et al. (2011) | ESM+chem+aero+icesheet | Interim |
| MRI-ESM2.0 | 6 | MRI-AGCM3.5, 320 × 160, TL159L80, 80 lv | MRICOM-4-4, 364 × 360, tripolar primarily 0.5 × 1.0° grid, 61 lv | r1i1p1f1 + 4 | Yukimoto et al. (2019) | ESM+chem+aero | Interim |
| NESM3 | 6 | ECHAM v6.3, 192 × 96, T63L47, 47 lv; | NEMO3.4-ORCA1, 46 lv | r1i1p1f1 + 4 | Cao et al. (2018) | AOGCM | none |
| NorESM1-M | 6 | CAM4-Oslo, 144 × 96, f19L26, 26 lv; | MICOM-noresm-ver1-gx1v6, 384 × 320, 53 lv | r1i1p1 | Bentsen et al. (2013) | AOGCM+aer | JRA-55 |
| NorESM2-LM | 6 | CAM-Oslo, 144 × 96, 32 lv; | MICOM, 384 × 360, 1.0° along the equator, 70 lv | r1i1p1f1 + 2 | Seland et al. (2020) | ESM+chem+aero+icesheet | mixed |
| NorESM2-MM | 6 | CAM-Oslo, 288 × 192, 32 lv; | MICOM, 384 × 360, 1.0° along the equator, 70 lv | r1i1p1f1 + 1 | Seland et al. (2020) | ESM+chem+aero+icesheet | Interim |
| SAM0-UNICON | 6 | CAM5.3 with UNICON, 288 × 192, 30 lv | POP2D, 320 × 384, 60 levels | r1i1p1f1 | Park et al. (2019) | AOGCM+aero | Interim |

**Table 1.** Overview of the applied model experiments, including the acronyms of the coupled models and their atmosphere and ocean components, the nominal resolution in lon × lat, the number of vertical model levels (lv), the run identifiers (complemented by Figure 11 for more than 1 run), reference articles, model complexity and reanalysis affinity (see text for details); the ocean meshes are defined as follows: ORCA2 = 182 × 149, 2° with meridional refinement to 0.5° near the equator; ORCA1 = 362 × 292, 1° with meridional refinement to $\frac{1}{3}$° near the equator; ORCA05 = 722 × 511, 0.5° with no refinement; ORCA025 = 1442 × 1050, 0.25° with no refinement; eORCA1.3 = 362 × 332, 1° with meridional refinement to $\frac{1}{3}$° near the equator; eORCA1 = 360 × 330, 1° with meridional refinement to $\frac{1}{3}$° near the equator; eORCA025 = 1440 × 1205, 0.25° with no refinement.





**Table 2.** Rank correlation coefficients between the median MAE values of the 46 models and various resolution parameters of the atmosphere or/and ocean component models. A significant relationship is indicated by an asterisk ($\alpha = 0.01$, two-tailed t-test, $H_0$ = zero correlation). See text for more details.

| Realm | Zonal | Merdional | Vertical | 2D | 3D |
|---|---|---|---|---|---|
| atmosphere | -0.63* | -0.65* | -0.21 | -0.64* | -0.65* |
| ocean | - | - | -0.38* | -0.39* | -0.45* |
| atmosphere + ocean | - | - | - | - | -0.55* |