# Peer review of "A circulation-based performance atlas of the CMIP5 and 6 models for regional climate studies in the northern hemisphere mid-to-high latitudes"

_Geoscientific Model Development, 2020_

## Author Comment (AC1)

**Author Response to Referee 1**

Dear Referee 1,

Thank you very much for taking the time to review this manuscript and for your valuable corrections and suggestions.

Your comment: *The paper provides an interesting and highly relevant analysis of CMIP5 and CMIP6 models with respect to the representation of circulation in the northern hemisphere. It also shows the general improvement from CMIP5 to CMIP6 in this aspect. The analysis criteria are especially interesting for e.g. the regional climate modelling community by having an additional evaluation criteria to the commonly used temperature and precipitation analysis.*

*I recommend to accept the manuscript after taking some minor points into account.*

Response: Many thanks for your interest in the study and for your positive feedback. For the revised manuscript, 10 additional GCMs and 2 additional members of CNRM-CM6-1 have been added to the evaluation, although this was not requested by any of the referees, making it even more exhaustive. Also, with the help of a small survey sent out to all modelling teams, the documentation about the components of the participating GCMs has been confirmed and further extended. Please find below a point-to-point list to your valuable comments and suggestions.

Your comment: *Abstract, line 2: In many applications relevant for decision making, and particularly when deriving future projections with the delta-change method, they are assumed to be perfect. --> Isn't the delta-change method rather assuming that the model biases are constant than assuming that models are perfect?*

Response: I have been thinking quite a bit about this sentence as well. What I mean here is that stakeholders not familiar with climate science, and most importantly politicians, run the risk of using delta change estimates (or multi-model mean values thereof) as if they were deterministic predictions actually to occur in the future, and would then base their decision making and ultimately legislation on this premise. GCM errors and the stationarity assumption you mention are technical issues stakeholders are normally not aware of. A solution on how such technical questions should influence practical decision making is difficult. However, there is no need for lengthy discussions in the abstract and, following your advice, I have downweighted and simplified this sentence to: "In most applications relevant for decision making, they are assumed to provide a plausible range of possible future climate states." (see lines 2-3 in the revised manuscript)

Your comment: *Line 8: Both approaches, however, are in principle unable to correct errors resulting from a wrong representation of the large-scale circulation in the global model. --> Dynamical downscaling, at least to some extent within their regional domain, can correct errors in the large-scale circulation.*

Response: Following your advice, this passage reads as follows in the revised manuscript (see lines 8-9): "For both approaches, however, it is difficult to correct errors resulting from a wrong representation of the large-scale circulation in the global model."

Your comment: *Line 14: The latest model generation --> add (CMIP6).*

Response: "(CMIP6)" has been added here

Your comment: *Introduction, line 50: they do not correct errors inherited from a wrong representation of the large-scale atmospheric circulation --> As already stated above, I think this is a bit too strongly formulated. I'd rather say "correction of errors inherited from a wrong representation of the large-scale atmospheric circulation is challenging".*

*Response:* You are right, this sentence now reads as follows: "Now while downscaling methods are able to imprint the effects of the local climate factors on the coarse resolution GCM, the correction of errors inherited from a wrong representation of the large-scale atmospheric circulation is challenging (Prein et al., 2019)" (see lines 49-51 of the revised manuscript)

Your comment: *Line 70: the three aforementioned regions --> Which regions are you referring to?*

Response: Here I refer to Greenland and the surrounding seas, the southwestern U.S. and the Gobi desert. For the revised manuscript, this sentence was removed from the Introduction section.

Your comments:

*Applied Data and Usage: Line 88: integrations for given model --> integrations for **a** given model*

*Line 101: and the considerations of other model developers --> and the considerations of other model **developments**.*

*Line 104: metadata provided the model output files --> metadata provided **by** the model output files.*

Line 111: but also the by the -> but also by the

Line 118: Roberts et al. (2019)) --> Roberts et al., 2019)

Methods: Line 196: being the the standard --> being the standard

Response: Thanks for careful reading, all these errors have been corrected in the revised manuscript.

Your comment: *Line 198: Is CRMSE used for the ranking as well?*

Response: The CRMSE is used here instead of the MAE since the original version of the Taylor-Diagram works with anomaly fields, i.e. removes the pattern mean value from observations and model data prior to calculating the error statistics (Taylor 2001).

Your comment: *Model contributions from ...: (This is a very useful overview!). Considering the EC-EARTH model: Do you think the good performance can be explained by its relationship to the ERA5 reanalysis in terms of model parts? Maybe it's worth adding a note on that. When you compare to JRA-55 you see that the performance of EC-EARTH drops (but it still outperforms many other models). Maybe this can also explain the additional outliers mentioned in line 575.*

Response: EC-Earth's atmospheric component was derived from ECMWF's Integrated Forecasting System, which was also used to produce the ERA-Interim reanalysis (ERA5 is not used in the present study). This might explain why the performance for EC-Earth is slightly better when compared with ERA-Interim instead of JRA-55. However, this effect is small and notably shifts in the model ranks only in those regions where the two reanalyses substantially differ from each other. In fact, the outliers you mention are mainly located in these 3 regions. As mentioned by you, the overall results depicted in Figure 11 do not change if JRA-55 is used as reference reanalysis instead of ERA-Interim. This is pointed out in lines 637-42 of the revised manuscript and visualized in the supplementary material (see figs-refjra55/as-figure-11-but-wrt-jra55.pdf therein).

Your comments: *line 512: not argument --> not **an** argument; line 520: it had to excluded --> it had to **be** excluded; line 582: to obtain the size of combined --> to obtain the size of **the** combined; line 604: been run been to --> been run to*

*Response:* Thanks for careful reading. The aforementioned text passages were removed or corrected in the revised manuscript.

Your comment: *Summary, discussion and conclusions, line 671: Select the most favourable model --> Although the proposed method is objective, I don't think it will allow the user to select "the most favourable model". First of all, it only covers a certain aspect (representation of circulation frequencies), and taking other performance scores into account (e.g. temperature biases) will give a different model ranking. Further, the ranking provided is based on annual frequencies. Looking at seasonal frequencies will probably also provide different rankings. So in the end, the selection of "the most favourable model" will be a subjective user decision depending on the weight he gives on different aspects. In summary, the performance atlas provided in the paper provides a very useful **additional** source for model selection, but will not provide a singular basis for that decision.*

Response: The applied methods surely only cover certain aspects of model behaviour. However, as you state below, since the Lamb Weather Types are well known to be associated with typical regional precipitation, temperature and wind patterns, they constitute a good overarching concept. The text passage you mention above no longer appears in the revised manuscript.

Your comment: *General: As far as I understood, the LWT classification only takes pressure gradients into account. Did you also look at biases in pressure, e.g. the monthly SLP pressure bias in the models? Is there a relationship between the ranking you calculate and the pressure bias, e.g. models with a large pressure bias perform not well. Or is it possible that models with a large pressure bias nevertheless show a good representation of LWT patterns?*

Response: From the results of many previous studies, I would say yes, there is a relationship, but I did not specifically assess this issue in the context of the present study. I would expect only a weak relationship between the bias of to the point-wise mean SLP and the MAE of the LWT frequencies because LWTs are defined by pressure gradients rather than absolute values. However, I might of course be wrong and it would be worthwhile to look into these relationships in the future, also in regard with temperature and precipitation biases.

Once again, many thanks for your valuable comments and suggestions and for your efforts to improve the manuscript.

---

## Author Comment (AC2)

**Author Response to Referee 2**

Dear Referee 2,

Thank you very much for taking the time to review this manuscript and for your efforts to further improve it. Please find below a point-by-point response to your valuable comments.

Your comment: *The paper "A circulation-based atlas of the CMIP5 and 6 models for regional climate studies in the northern hemisphere" by Swen Brands analysis regional atmospheric circulation patterns based on Lamb (1972) for a set of CMIP5 and CMIP6 models. In general, the paper is a rich information source and an important contribution to the climate science community. In my view there are only some minor points that could be addressed differently or better.*

Response: Many thanks for taking the time to review this study and for your positive feedback. Please find below a point-to-point list of responses to your valuable comments.

Your comment: *In section 6 it might be worth mentioning that this study can serve as a basis for studies on future changes in LWT. It could be interesting to investigate, e.g. if certain LWT changes can be traced back to model families or model configurations (AOGCMs vs. ESMs).*

Response: You are right, this interesting idea is mentioned in more general terms in lines 743-745 of the revised manuscript.

Your comment: *Lines 575-594: The explanation of Table 2 and the metric behind is a bit too brief in my view. It seems an interesting analysis, but I cannot fully follow the different steps resulting in the numbers of Table 2. A bit more detailed explanation here, would be good.*

Response: Thanks for your valuable comment. In the revised manuscript, the procedure used to obtain the different mesh sizes is now explained with more detail (see lines 643-653) and an additional boxplot is introduced to visualize the relationship between the horizontal mesh size of the atmospheric model components and the coupled models' performance (see Figure 13, panel a).

Your comment: *Lines 621-622: I would just drop the name Commonwealth models here. It is used only once thereafter in the following sentence and does not improve the text (it might even confuse more than it helps).*

Response: The "Commonwealth" term has been completely removed from the revised manuscript.

---

## Author Comment (AC3)

**Author response to Dr. Roland Séférian**

Dear Dr. Séférian,

Thank you very much for taking the time to review the present study. Please find below a point-to-point list of responses to your very valuable comments and suggestions.

Your comment: *This work provides a relevant and timely analysis of CMIP5 and CMIP6 models with respect to the representation of circulation in the northern hemisphere. On top this work goes well beyond a routine assessment of global model performance because some of those global models will be used to drive lateral boundary conditions of regional models or to derive climatic impact-drivers at regional/local scale. While it also shows the general improvement from CMIP5 to CMIP6 in this aspect, this work shine light on some deficiency within the current generation of models. With the objective the author tries to map model performance on two axes: the complexity and the resolution, both of which are difficult to separate.*

Response: Thank you very much for your interpretation of the study. The principal aim of the study is to provide a performance estimate for the GCM configurations participating in CMIP5 and 6, which is based on recurrent regional atmospheric circulation patterns as suggested in Maraun et al. (2017). The secondary aim is to provide a simple approach to measure the complexity of these models, which might then be used as additional GCM selection criterion apart from model performance (see Section 3.3, Table 1 and Figure 13b). To my knowledge, this point has been rarely taken into account in regional climate studies so far. The approach proposed here should be interpreted as a reasonable starting point to measure model complexity and is, of course, open to further modifications and specifications in the future. A corresponding Python function is available at github and can be edited by everyone interested to do so. Importantly, the present study is only *one* among many other studies currently taken into account for GCM selection in regional climate initiatives.

Your comment: *In consequence, the high-level picture of the analysis emerging for this work strongly tights to the Table 1 — where we spotted some errors. For instance, it is indicated that CNRM-CM6-1 and CNRM-CM6-1-HR included online chemistry onboard whereas the description of these model configurations in Voldoire et al. (2019) doesn't support this feature. Same goes, for IPSL models and for GFDL-CM4 which are characterized as 'ESMs' in Table 1 while they do not fit the current understanding of what is an Earth system models (see Jones (2019)). As shown in Séférian et al. (2020), GFDL-CM4 indeed included marine biogeochemistry but only in a stylized manner (reduced complexity marine biogeochemical models). In consequence, there are no biophysical feedbacks represented in GFDL-CM4 whereas it does in GFDL-ESM4.*

Response: Many thanks for revising Table 1, I very much appreciate your comments on this table, since it is key for the understanding of the study. Regarding CNRM-CM6-1 and CNRM-CM6-1-HR, an interactive atmospheric chemistry model was erroneously added in this table because the source attribute of the netCDF output reads as follows (the following example is for CNRM-CM6-1):

u'CNRM-CM6-1 (2017): aerosol: prescribed monthly fields computed by TACTIC_v2 scheme atmos: Arpege 6.3 (T127; Gaussian Reduced with 24572 grid points in total distributed over 128 latitude circles (with 256 grid points per latitude circle between 30degN and 30degS reducing to 20 grid points per latitude circle at 88.9degN and 88.9degS); 91 levels; top level 78.4 km) atmosChem: **OZL_v2** land: Surfex 8.0c ocean: Nemo 3.6 (eORCA1, tripolar primarily 1deg; 362 x 294 longitude/latitude; 75 levels; top grid cell 0-1 m) seaIce: Gelato 6.1'

Since "atmosChem: OZL_v2", I interpreted this as an interactive component model, as is normally the case if a model is specified for a given realm and the term "prescribed" is missing (compare with the *atmosChem* entry with the the *aerosol* entry above). I had noticed that this was in disagreement with Voldoire et al. (2019), but unfortunately gave preference to the aforementioned source attribute which I interpreted wrongly.

Many thanks also for pointing out that ocean biogeochemistry in GFDL-CM4 is represented by a reduced complexity marine biogeochemical model without biophysical feedbacks. In the revised manuscript, the respective complexity integer was consequently set to 1.

In order to avoid such errors, the complexity codes provided in Table 1, column 7 of the revised manuscript have been confirmed and corrected by the corresponding modelling teams by means of e-mail correspondence.

Considering Collins et al. (2011), Jones (2020) and personal e-mail correspondence I have had with two modelling groups, I have come to the conclusion that there exist at least 3 different definitions for the term "Earth System Model":

1. "ESM could also be defined as adding other than pure physical processes of ocean and atmosphere to the classical GCM." (personal communication with the EC-Earth group)

which is qualitatively identical to:

"These models are now known as Earth System Models (ESMs) to denote that they simulate more than just the "physical" elements of the world's weather" (Jones 2020, page 1)

2. "There is no strict definition of which processes at what level of complexity are required before a climate model becomes an Earth system model [...] however typically the term "Earth system" is used for those models that at least include terrestrial and ocean carbon cycles." (Collins et al. 2011, page 1051, this definition was used in the first version of the manuscript)

3. The ESM term is "mostly about fashion, less about content" (personal communication with a developer from one important GCM group)

Since the ESM term is not clearly defined, it is avoided in the revised manuscript. Instead, the manuscript now simply refers to "more" or "less" complex models.

Your comment: *Regarding the axis of the resolution, providing the nominal resolution would help to compare model between each other. The nominal resolution has been reported by the modelling groups to CMIP6 for each component/realm.*

Response: The nominal resolution reported to CMIP6 is only approximate and this is why the present study works with the mesh size of the atmosphere and ocean grids, as indicated in the source attributes or directly by the data arrays in the netCDF files. This approach is also approximate, but likely more exact than reported nominal resolutions. Also, the nominal resolution of the model versions participating in CMIP5 has not been reported, which is another reason for the use of the alternative approach.

Your comment: *Apart from these remarks/on Table 1, we would like to provide a couple of suggestions that could be useful for this work.*

*As this work focus on the performance over the historical period, it might be relevant to provide some information on how the model has been tuned/calibrated. At least to know if this set of metrics has been used as a target to prepare the model for CMIP5 and for CMIP6. Such questions tend to emerge now in the literature (see Spafford and MacDougall, in review ni GMDD) because of their implication on routine performance benchmark.*

Response: This is a very interesting issue, which however is very difficult to trace back to all model configurations used in the present study, inlcuding those from CMIP5. For the model family performing best here (EC-Earth3), Klaus Wyser and Ralf Döscher confirmed that: "In the EC-Earth3 tuning process, regional SLP patterns were not a target." via e-mail correspondence. A further look at Döscher et al. (2021) reveals that "The atmospheric component of EC-Earth has been tuned with the goal of achieving a reasonably small radiative imbalance at the top of the atmosphere". A more systematic assessment of this issue is interesting, but out of the scope of the present study.

Your comment: *On the other hand, the paper is not clear on the treatment of the model realization. As shown in Olonscheck et al (2020), large ensemble of realization may improve the comparison with the observation. Considering the magnitude of the internal variability of the atmospheric circulation feature, considering additional information on available model member might help. With that said, comparing model with different ensemble size might complicate the picture but discussing the impact of the member on the overall model performance and ranking would be a very valuable outcome of the paper.*

Response: I am afraid that this might be a misunderstanding since already in the first version of the manuscript a total of 70 alternative runs from 12 distinct GCMs were assessed to estimate the role of internal variability (see Figure 11 in manuscript version 1). This long list has now been extended to 72 alternative runs from 13 distinct GCMs. Namely, 2 additional CNRM-CM6-1 members were included in Figure 12 of the revised manuscript. As you can see from these figures, the effect of internal variability on the overall results is negligible.

Your comment: We hope that the author will find these comments and suggestions useful/relevant.

Response: I am very grateful to your valuable corrections and suggestions. Thank you very much for taking the time to review this study.

The references cited in this response letter are listed in the revised manuscript.

---

## Author Response (AR2)

**Response to referee 3**

Dear Dr. Añel,

Thank you very much for supervising the peer-review process of the present manuscript and for your additional referee services. Please find below a point-by-point response list to your valuable comments.

Your comment:
*A few issues remain in your manuscript, and I think that need to be addressed. At least they will help to make more straightforward the message of your work.*

*The first one is about the title. Currently, it reads, "A circulation-based performance atlas of the CMIP5 and 6 models for regional climate studies in the northern hemisphere". The first word that does not work in this title is "atlas". From the very beginning, I was expecting to see cartographic plots. Indeed, this is what most people think when they read "atlas". However, your work is an evaluation. Therefore, I believe that you should change this word to "analysis" or "evaluation". The second issue is that you state that the evaluation is for the northern hemisphere, but this is not true. It is precisely for mid-latitudes, as you clarify later in the paper. Therefore, I think that a title that translates better the contents of your work would be: "A circulation-based performance analysis of the CMIP5 and 6 models for regional climate studies in the northern hemisphere mid-latitudes". I recommend you modify it accordingly.*

Response:
I fully agree with you that the study region should be specified with more detail in the title. I would, however, prefer to refer to "mid-to-high" latitudes as initially proposed by Jones et al. (1993, page 1129). They state that the Lamb weather typing approach can be used anywhere in between 30º-70º and refer to this as "mid-to-high" latitudes. In the present study, the method is applied at 35-70ºN, which is why I would like to use the aforementioned nomenclature. Please note that the Lamb coordinate system centered at 70ºN actually extends to 80ºN, i.e. well into the high latitudes. Since figures 2 to 10 all together provide more than 100 maps and the region-specific evaluation is an important part of the main results (see Sections 4.1 to 4.8), it is fully justified to use the term "atlas" in the title from my point of view. I have therefore changed it as follows:

"A circulation-based performance atlas of the CMIP5 and 6 models for regional climate studies in the northern hemisphere mid-to-high latitudes"

However, this is of course not a critical issue and I could switch to "analysis" or "assessement" if this is requested by you.

Your comment:
*The Abstract must not contain citations. Please, remove them.*

Response: Thank you very much for pointing this out. I have removed all citations from the abstract.

Your comment:
*You continue mentioning a Github repository, which is mentioned both in the main text and the Code availability section. And it does not make sense because the "get_historical_metadata.py" function is already stored in the Zenodo repository, which confuses the reader. When mentioning the function in the text, please state that it is available in Zenodo, not Github. And in the Code availability section, simply remove the mention to Github.*

Response:
As suggested by you, I have redirected all links related to my work to the respective Zenodo entries. Some applied Python packages have not been permanently stored on Zenodo or similar alternative archives and this is why the links to the respective github repositories have not been removed (see Section 3.4).

Your comment:
*Section 4: This section is entitled 'Overall model performance results', but subsections 4.1-4.8 are merely descriptions of the models. This information takes the following nine pages and avoids focusing on comparisons. I know it is relevant to understand potential differences between models; however, it would better fit an Appendix. Therefore, please, see our guidelines for Appendices and move these sections there. You can retain in the main text the few mentions to the conclusions obtained, citing the Appendix for additional information on the models.*

Response: I am afraid this might be a misunderstanding. "Overall" here refers to the performance for all 27 circulation types, as opposed to the type-specific evaluation in Section 5. Sections 4.1 to 4.8, apart from the model descriptions, indeed also contain the main results. Model performance maps for the entire northern hemisphere mid-to-high latitudes are provided and described there, meaning that these sections are indispensable for a proper understanding of the models' performance for particular regions, which is currently demanded by the dynamcial and statistical downscaling community. Thus, I would not like to move sections 4.1 and 4.8 to the appendix.

I have been thinking about moving only the model descriptions to the appendix. However, as mentioned by you, this would disrupt the argumentation line. Similar or equal atmospheric models return similar spatial ranking patterns and this is why the model descriptions are placed in the main text. Note also that the descriptions in the main text focus on the atmosphere, land-surface, ocean and sea-ice models, meaning that they have been already reduced. The full descriptions including up to 6 additional realms are provided in *get_historical_metadata.py*. The condensed model descriptions in the main text are very usefull since they can be used by regional climate modelers (CORDEX) to select those global models that are largely independent in order to generate ensemble spread. Finally, in the first revision round, referee 1 praised the "Model contributions from…." sections by stating "This is a very useful overview!" and this is why I would prefer to leave them as they are. If the manuscript is too long, I would rather prefer to move figures 14, 15 and 16 to the supplementary material. These roughly occupy the same space then the model descriptions in Sections 4.1 to 4.8.

Your comment:
*In lines 272 and 603, you mention Figure 11. However, these citations are unnecessary, and you cite Figure 11 after barely saying Figs. 3-10, which are only mentioned for evaluation purposes. At least in line 272, it is unnecessary to cite it. Please, remove it.*

Response: The citation of Figure 11 in line 272 has been removed, as suggested by you.

Your comment:
*Line 485: Provide only information, not opinions. For example: "is another example for the success of long-standing research efforts from many research institutes around the world" is a particular view that does not add anything to an already lengthy manuscript. Please, double-check the text and remove this kind of statement.*

Response: You are perfectly right. This is a personal view which has been removed from the revised manuscript. The manuscript has also been revised in this respect, as suggested by you.

Your comment:
*The "complexity" issue: I have read this part of the manuscript with interest. Complexity is a polysemic word, and indeed, you use it in different ways along the manuscript. For example, in line 40, it seems to have a broader meaning than the one you introduce later, mainly dealing with the number and coupling of submodels that run in a model. In this way, model "complexity" can simply measure the number of lines of code and subprocesses.*

Response:
Thank you very much for discussing the complexity codes provided in the study. In the former version of the manuscript, the term "complexity" is defined at first glance in the abstract (lines 18 – 20) as "a relatively simple approach based on the number of climate system components taken into account by the GCMs" and is later on described with detail in Section 3.3, so its meaning should be clear. However, I agree that some phrases in the former version of the manuscript, and particular the usage in line 40, might be interpreted otherwise by some readers. In the revised manuscript I am submitting now (version 4), I have carefully checked the text of the entire manuscript in order to avoid such misunderstandings. The usage of the term "complexity" in the present study is clearly defined in Section 3.3, now entitled "Model complexity in terms of considered climate system components" (see lines 190 - 195 of the revised manuscript). The limitations of the approach are clearly stated and some interpretation guidelines are provided in lines 210 - 216, so there is no room for misunderstanding any more.

Defining model complexity by the number of code lines and subprocesses is an interesting alternative approach and, at least in theory, is more exact than the approach followed in the present study. However, as is pointed out in Añel et al. (2021), the source code of many CMIP models is simply not available and this is why the approach suggested by you is very difficult, if not impossible, to achieve in practice, particularly when a large number of GCMs are assessed, as is the case in the present study. Likewise, even if the the source codes for all these models were available, you would first have to know which model components were actually activated for the historical experiments run for CMIP5 and 6, and this is reflected by the complexity code introduced in the present study. As such, this study may be interpreted as a necessary prequisist for a full assessement of the source code, which however mains elusive due aforementioned access restrictions. Also, the use of different programming languages or versions thereof and different code efficencies would

hamper a comparison among different models. From my point of view, the only feasilbe way to achieve a fully comprehensive complexity assessment would be a large community effort involving the code developers themselves, with one paper per climate system component, following the example of Séférian et al. (2020) for the case of ocean biogeochemistry. Such an assessement is clearly out of the scope of the present study.

Your comment:
*I'm not at all convinced that a model is more complex by the simple fact of having an interactive module or not. A simple atmospheric model with many processes described (e.g., gravity waves, spontaneous QBO, resolved stratosphere, interactive chemistry) can be much more intricate than others coupled to an ocean but with processes parameterized.*

Response:
The 56 model versions used in the present study are all from the same class of *fully comprehensive coupled general circulation models*, meaning that the complexity of the component models for a given realm (atmoshere, land-surface, ocean, sea-ice etc.) should be roughly comparable from one coupled model configuration to another and the inclusion of additional realms should be proportional to an increase in the number of represented processes and code lines. This is particularly the case for the coupled model configurations of the same family (e.g. EC-Earth2.3, EC-Earth3, EC-Earth3-Veg, EC-Earth3-AerChem and EC-Earth-CC) whose component models often differ in the version number only. Furthermore, *nominally* distinct coupled model families often share their submodels (or version thereof) for one or several realms, in which case they are identical or very similar in these realms. Namely, the 56 nominally different coupled model configurations considered here only use 19 different atmosphere general circulation models or versions thereof and the number of independent ocean models is even lower (see Table 1, columns 3 and 4). Therefore, I would expect the approach followed here to provide reasonable *estimates* for a fully comprehensive model complexity assessement based on source code as suggested by you. However, I do not mention this any more in the revised manuscript because it cannot be proven to date.

Likewise, I have never attempted to provide a full complexity assessment and do not claim either that the approach presented in the present paper can be used for this purpose. Consequently, it is not necessary to prove this even if the source code for all GCMs was available.

To ensure the validity of the simplified approach followed in the present study, I designed a survey and sent it by e-mail to up to three scientists of each modelling group. The survey contained a description of the approach which explicilty stated that "model complexity" is *estimated*, as well as an initial complexity code proposal based on profound reading of the reference articles and metadata contained in the netCDF files from ESGF (the survey is provided further below in this response letter). Out of the 19 contacted modelling groups, 17 confirmed or corrected the code and 2 did not answer. Among the 17 groups providing feedback, only a single scientist from one group stated that he/she is not sure whether the proposed approach is suitable to measure model complexity, but did not reject it either. In light of the many contacted scientists, this is a very favourable result. On demand, I can share the entire e-mail correspondence I have had with the 19 modelling groups with you.

Note also that I designed the simplified complexity approach in response to the community comment posted by Roland Séférian in the first revision round. This was done in order to find a flexible solution for a definition of the term "Earth System Model", for which no consensus exists

between the different modelling groups (see my response letters from the first revision round). The aforementioned survey was also sent to Roland Séférian, Aurore Voldoire and Samuel Somot from CNRM. Aurore Voldoire responded as the group leader (CC-ing Roland Séférian and Samuel Somot). She corrected the complexity code, made some clarifications about the atmospheric chemistry module used in the CNRM models, which are stored in get_historical_metadata.py, and did not mention any doubts about the approach itself. If Roland Séférian had a major problem with the approach, he would have surely informed me on this occasion.

Your comment:
*Moreover, your explanation of how you compute the complexity number for each model is incomplete and hard to decode from Table 1.*

Response: In the former version of the manuscript (the one you refer to), the complexity code was restricted to the 6 components 1. Vegetation properties, 2. Terrestrial carbon-cycle processes, 3. Aerosols, 4. Atmospheric Chemistry, 5. Ocean biogeochemistry and 6. Ice sheet dynamics because the four basic components atmosphere, land-surface, ocean and sea-ice are interactive in all participating models and thus do not contribute variability to the complexity code. For ease of understanding, and also to be consistent with the use of the code in the EURO-CORDEX initiative (see link below), all 10 realms contribute to the complexity code in the revised manuscript, which now comprises 10 instead of 6 integers for each GCM version: 1. Atmosphere, 2. Land-surface, 3. Ocean, 4. Sea-ice, 5. Vegetation properties, 6. Terrestrial carbon-cycle processes, 7. Aerosols, 8. Atmospheric Chemistry, 9. Ocean biogeochemistry and 10. Ice sheet dynamics.

Your comment:
*In the end, your proposed code does not seem to add too much to the discussion, as you state in the Conclusions. Only a few apparent discrepancies between less complexity and better performance arise.*

Response: I do not agree on the point that the proposed complexity estimate does not add too much to the discussion and do not state this in the conclusions either. Indeed, the complexity scores depicted along the x-axis of Figure 13b are higher for ACCESS-ESM1.5, CMCC-ESM2, CNRM-ESM2-1, EC-Earth-CC, MIROC-ESM and HadGEM2-ES than for the remaining model versions of the same modelling group, which is in line with the corresponding reference articles and shows that the method yields reasonable results in spite of its relative simplicity. The complexity approach used in the present study is also currently taken into account by the model selection team of the EURO-CORDEX community. You can contact Dr. Jesús Fernández (person of contact in EURO-CORDEX) on this issue or have a look at the github entry related to GCM evaluation for EURO-CORDEX, which is located at [https://github.com/jesusff/cmip6-for-cordex/blob/main/CMIP6_studies/Bra21.yaml](https://github.com/jesusff/cmip6-for-cordex/blob/main/CMIP6_studies/Bra21.yaml)

Your comment:
*It is hard (if not doubtful) to assess how complex is a model without checking its source code.*
*Entire papers are devoted to this task that you intend to do here in only twenty-five lines of*
*explanation. Therefore, given all this reasoning, I suggest that you remove this subsection and any*
*mention of this metric from the manuscript.*

Response:
I agree that an analysis of the model source codes would probably yield more exact results in this
regard. However, as pointed out in Añel et al. (2021), such an analysis is not feasible in practice
since the source code for many of the 56 model versions participating here is not available, apart
from other problems mentioned above.

I also agree on the fact that entire papers on this task exist (e.g. the impressive work of Séférian et
al. 2020), but these are very profound analyses of *single* climate system components (ocean
biogeochemistry in the latter case) conducted for a lower number of GCMs. Since the approach
followed in the present study covers 10 climate system components and a very large number of
coupled model versions (56), it is broader and necessarily less profound than the aforementioned in-
depth studies. This was recognized in the former manuscript version (in lines 744-747) and is
emphasized in the present one (in lines 210-216), leaving no room for misunderstandings. As
mentioned above, the approach followed here resides on profound reading of the reference articles,
analysis of the metadata within the netCDF files from ESGF and, most importantly, on the approval
of the global model developers themselves, and is thus valid. It is also being used by the model
selection team of the EURO-CORDEX initiative in their effort to downscale CMIP6 simulations,
showing its timeliness and relevance.

Given these arguments, there is no reason to remove the approach from the manuscript. However, I
do understand your point. I think we are having a simple nomenclature problem that leads to
circular reasoning. In the present study, I have defined a code reflecting the number of prescribed
and interactively simulated climate system components for a large number coupled model
simulation from CMIP and refer to this as "model complexity". You state that model complexity is
something else, namely the number of code lines and subprocesses included in the coupled model
configurations. To avoid this nomenclature problem, I would be willing to replace the term "model
complexity" by another expression like "model comprehensiveness", "degree of interactivity" or
"climate sytem representativity", if this is requested by you. From my point of view, however, this
is not necessary because the way the term "model complexity" is used in the present study has been
approved by nearly all modelling groups in the aforementioned survey, and has been rejected by
*none* of them. This is in line with the recommendation of referee 1, who has accepted the former
version of the manuscript (version 3) as is.
Finally, the fact that the approach is summarized in twenty-five lines does not reflect the real
underlying work, which was very extensive. This is documented by more than 1500 code lines
incorporated in *get_historical_metadata.py*, gathered manually on the basis of profound reading,
file analyses and a survey involving the modelling teams, whose help is recognized in the
Acknowledgements. In the revised manuscript, Section 3.3 has now been extended, including
details about the survey and its outcome.

I very much appreciate your efforts to further improve the manuscript and remain

with kind regards,

Swen Brands

**References**

Añel JA, García Rodríguez M, Rodeiro J (2021): Current status on the need for improved accessibility to climate models code, Geoscientific Model Development 14(2), 923-934, doi: 10.5194/gmd-14-923-2021.

Jones P D, Hulme M, and Briffa KR (1993): A comparison of Lamb circulation types with an objective classification scheme, International Journal of Climatology, 13, 655–663, doi:10.1002/joc.3370130606.

Séférian R, Berthet S, Yool A, Palmiéri J, Bopp L, Tagliabue A, Kwiatkowski L, Aumont O, Christian J, Dunne J, Gehlen M, Ilyina T, John J, Li H, Long M, Luo J, Nakano H, Romanou A, Schwinger J, Yamamoto A (2020): Tracking Improvement in Simulated Marine Biogeochemistry Between CMIP5 and CMIP6, Current Climate Change Reports 6, 95–119, doi: 10.1007/s406411085020-00160-0.

**Survey**

The survey sent by e-mail to each of the 19 modelling teams is as follows (this is the example sent to CNRM):

Subject: Question about the CNRM-CM model versions contributing to CMIP5 and 6

Dear Dr. Voldoire, Dear Dr. Séférian, Dear Dr. Somot,

In an attempt to create an inventory about the complexity of the *historical* coupled general circulation model (GCM) experiments submitted to CMIP5 and 6, I would kindly ask for your help. I am including Dr. Somot because we possibly will consider GCM complexity within the EURO-CORDEX initiative.

I would like to find out how the distinct components of the climate system are represented in these experiments. For the CMIP6 runs, this is reflected in the "source" attribute within the netCDF output files distributed by the ESGF data portals. However, the source attribute does not provide information on whether a given component is interactive, prescribed, or something in-between (which is sometimes called "semi-interactive"). Within the inventory I am currently building, I take into account the following components:

1. atmosphere
2. land surface
3. ocean
4. sea-ice
5. dynamic vegetation (*this evolved to "vegetation properties" in later phases of the survey*)
6. terrestrial carbon cycle processes
7. aerosols
8. atmospheric chemistry
9. ocean biogeochemistry (ocean carbon cycle processes)
10. ice sheet dynamics (Greenland and Antarctic ice sheets)

I assign an integer 0, 1 or 2 to each of these components in case they are absent (0), prescribed or semi-interactive (1) or interactive (2). On the basis of the source attributes and reference articles, I come to the following integer combinations for the CNRM-CM model runs submitted to CMIP5 and 6:

CNRM-CM5, r1i1p1: 2222101101
CNRM-CM6-1, r1i1p1f2: 2222111101
CNRM-CM6-1-HR, r1i1p1f2, 2222111101
CNRM-ESM2-1, r1i1p1f2, 2222222221

If you could confirm and correct possible mistakes, that would be great. I am building the inventory as a simple Python function which also contains the names, resolution details and other information about the distinct component models. I will upload this function to GitHub and can include you as contributor if you like to. Many thanks for your help.

With kind regards,

Swen Brands